# Neural Fourier Modelling: A Highly Compact Approach to Time-Series Analysis

## Abstract

Neural time-series analysis has traditionally focused on modeling data in the time domain, often with some approaches incorporating equivalent Fourier domain representations as auxiliary spectral features. In this work, we shift the main focus to frequency representations, modeling time-series data fully and directly in the Fourier domain. We introduce Neural Fourier Modelling (NFM), a compact yet powerful solution for time-series analysis. NFM is grounded in two key properties of the Fourier transform (FT): (i) the ability to model finite-length time series as functions in the Fourier domain, treating them as continuous-time elements in function space, and (ii) the capacity for data manipulation (such as resampling and timespan extension) within the Fourier domain. We reinterpret Fourier-domain data manipulation as frequency extrapolation and interpolation, incorporating this as a core learning mechanism in NFM, applicable across various tasks. To support flexible frequency extension with spectral priors and effective modulation of frequency representations, we propose two learning modules: Learnable Frequency Tokens (LFT) and Implicit Neural Fourier Filters (INFF). These modules enable compact and expressive modeling in the Fourier domain. Extensive experiments demonstrate that NFM achieves state-of-the-art performance on a wide range of tasks (forecasting, anomaly detection, and classification), including challenging time-series scenarios with previously unseen sampling rates at test time. Moreover, NFM is highly compact, requiring fewer than **40K** parameters in each task, with time-series lengths ranging from 100 to 16K.

## 1 Introduction

Time series analysis is valuable in understanding the dynamics of systems and phenomena that evolve over time, and to address practical problems in a range of domains. With rapidly increasing computational resources and data available for learning, neural-based modelling approaches (Vaswani et al., 2017; Oord et al., 2016) have recently gained vast popularity in the discipline. A number of sophisticated methodologies and models have been developed, greatly advancing performance on a variety of time-series tasks such as forecasting (Nie et al., 2022; Wu et al., 2021), classification (Zhang et al., 2022; Raghu et al., 2023), anomaly detection (Xu et al., 2021a; Chen et al., 2022a). Behind this success, a remaining central question in time series modelling is how to capture meaningful information relevant to tasks from temporal patterns and generalize the dependencies ingrained

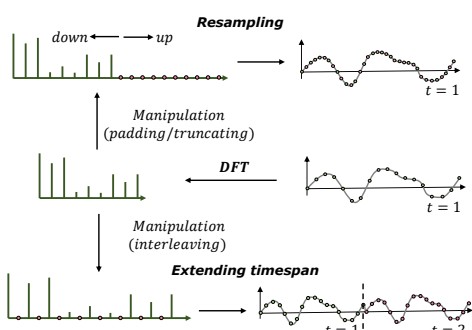

Figure 1: Illustration of Fourier-domain manipulations (left), including zero-padding/truncation (top) and zero-interleaving (bottom), and equivalent effects in the time domain (right).

within time-evolving data that are inherently diverse and intricate. To answer the question and thus progress the discipline further, in this work we study frequency representations, giving rise to a powerful time-series modelling scheme, Neural Fourier Modelling (NFM).

Given its well-known prevalence in the field of conventional signal processing, the adoption of frequency domain analysis to neural-based modelling is, unsurprisingly, not a unique idea in itself. There have been a number of research efforts across multiple areas from time-series analysis to computer vision (CV). Spectral representations are often harnessed as an alternative (Trabelsi et al., 2018; Choi et al., 2019) or as a complement (Yang & Hong, 2022; Woo et al., 2022) to time- or spatial-domain representations to capture information in a more compact and overarching form. Moreover, several recent studies have extended its applications to an efficient form of global convolution (Huang et al., 2023; Lee-Thorp et al., 2021; Lin et al., 2023; Alaa et al., 2020; Yi et al., 2024a) and a means of data augmentation (Xie et al., 2022; Xu et al., 2021b; Zhou et al., 2022b) for facilitating invariances conducive to generalization in learning. While the existing works have successfully utilized the frequency representation, it still remains under-explored with the lack of study on frequency interpolation and extrapolation as a direct means of modelling data fully in the Fourier domain. More specifically, our work is motivated by two aspects of frequency representations. (i) Simply learning directly in the Fourier domain enables finding a function-to-function mapping - an inductive bias towards learning resolution-invariance property. (ii) Fourier-domain manipulation and a fundamental connection to its time-domain counterparts can provide a general means of learning in the Fourier domain.

**Fourier-domain Manipulation.** As shown in **Figure 1**, there are two ways to manipulate time series in the Fourier domain – 1) *padding/truncating* and 2) *interleaving* the original Frequency representation with zero coefficients, each of which resulting in resampling and extending the original time-domain representation, respectively. Taking this into a view where a resultant time-domain representation caused by the frequency manipulation to a given input sequence is a desired target, we can naturally reformulate the manipulation with zero frequency coefficients into a constructive process of learning meaningful coefficients towards the target - i.e., *frequency extrapolation and interpolation*. Notably, this view provides a comprehensive learning framework requiring no architectural modification to models. For example, time series can be readily modelled for forecasting task from the frequency interpolation. Moreover, context learning (e.g., classification and regression) or representation learning can be made through the frequency extrapolation or directly imposing reconstruction practice with an auxiliary modelling scheme such as Fourier-domain (Xie et al., 2022; Zhang et al., 2022) masking.

Adopting the above insight as a core learning mechanism, we frame time series modelling into finding an interpolation or extrapolation solution between input and target directly in the Fourier domain and propose NFM. To achieve this, we introduce two main learning modules that operate directly in the Fourier domain and equip NFM with them. (i) A complex-valued learnable frequency token (LFT) is proposed to capture effective spectral priors and enable flexible frequency extension for the frequency interpolation and extrapolation. (ii) Implicit neural Fourier filter (INFF) as a principal processing operator is designed to realize an expressive continuous global convolution for learning the interpolation or extrapolation in the Fourier domain. We apply NFM to various datasets and distinct tasks of scenarios with both normal (constant) and unseen discretization rate, and show that NFM achieves state-of-the-art performance in a remarkably compact form - forecasting with **27K**, anomaly detection with **6.6K**, and classification with **37K** parameters.

## 2 RELATED WORK

**Frequency representations for time series modelling.** There has been a growing attention for processing time series and learning temporal dynamics of it with Fourier-domain information and/or through Fourier-domain operations in neural-based time series modelling methods. Autoformer (Wu et al., 2021), FEDformer (Zhou et al., 2022b), and FourierGNN (Yi et al., 2024a) adopt a frequency-based mixing mechanism as a main operator for learning temporal dependencies, where the computation efficiency is ensured by operating in the Fourier domain with the Fast Fourier Transform (FFT) algorithm. FiLM (Zhou et al., 2022a), BTSF (Yang & Hong, 2022), and TimesNet (Wu et al., 2022) leverage frequency representations in conjunction with time-domain representation to have a better realization of long-term dependency and global patterns (low frequency components). COST (Woo et al., 2021) and Autoformer exploit the Fourier domain as an explicit inductive bias, utilizing it not only for efficient computation but for decomposing periodic patterns from complex time series. TF-C (Zhang et al., 2022) enhances transferability of representations by integrating spectral

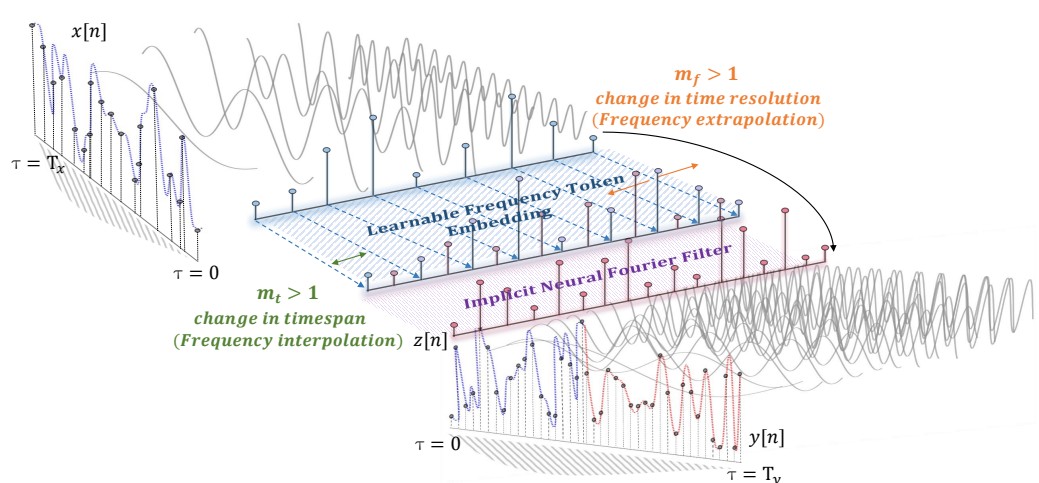

Figure 2: Overall workflow (forecasting scenario is exemplified) of the proposed NFM which deals with discrete signals as continuous-time elements in the compact function space through Fourier lens. NFM finds an interpolation/extrapolation from discrete input to target directly in the Fourier domain.

representations and introducing Fourier-domain augmentation in contrastive learning framework. The current utilization is capitalized on enabling efficient computation over long sequences and complementing the point-wise time-domain representations with the overarching representations. Differing from these works, Yi et al. (2024b) designs FreTS fully with Fourier operators and models time series fully in the Fourier domain for forecasting task. Our work follows in a similar vein to FreTS in the sense that we model time series directly in the Fourier domain and fully leverage on learning with the samples of functional representations of discrete signals, but with the distinct difference in the used core learning mechanism, frequency extrapolation/interpolation.

**FITS.** More recently, Frequency Interpolation Time Series analysis baseline (FITS) (Xu et al., 2023), a remarkably lightweight frequency-domain linear model, is introduced to address time-series problems. While NFM is designed leveraging the analogous principle (frequency interpolation and extrapolation) as FITS, there are several improvements essential for practicality - see details in **Appendix A**. In short, NFM is a generalization of FITS, that can 1) model both multivariate and univariate time series, 2) readily scale up, 3) be adaptive to variable-length inputs/outputs, and 4) be as compact as FITS and even surpass FITS's compactness while yielding better performance.

## 3  NEURAL FOURIER MODELLING

In this section, we provide an overview of NFM (**Section 3.1**) and introduce two main learning modules, Learnable Frequency Tokens (LFT) (**Section 3.2**) and Implicit Neural Fourier Filter (INFF) (**Section 3.3**). To begin, we first provide notations and necessary preliminaries below.

**Notations.** Considering a $c$-channel signal $x \in (\mathcal{D}, \mathbb{R}^c)$ and a target function $y \in (\mathcal{D}, \mathbb{R}^{d_y})$ defined on some temporal domain $\mathcal{D} \subset \mathbb{R}$, let $D_i = (x_i, y_i) \subset \mathcal{D}$ be a $i$th pair of input signal and target in domain $\mathcal{D}$. We denote $I_O = \{0, ..., O-1\}$ a set of indices for integer $O \geq 1$ and define $x[n \in I_N]|_{D_i} = x(n/f_x)|_{D_i}$ as a time series with $N$-point discretization at rate $f_x$ over the timespan $[0, T_x]$, i.e., $N := T_x f_x$. The target $y$ can be in any form, depending on tasks (see **Appendix E**), and its information spanning on its $L$-point latent representation $z[n \in I_L]|_{D_i}$ over an output timespan $[0, T_y(\geq T_x))$ at sampling rate $f_y$, where $L(\geq N) := T_y f_y$. For notational simplicity, we drop $|_{D_i}$ and set $T_x$ to a unit timespan and subsequently the input sampling rate $f_x = N$.

**Preliminary: Relationship between input and output discretization.** In our framework, it is important to pay an attention to the relationship between the input and output discretization $N$ and $L$ as it removes ambiguities in formulating time series problems. For example, a task with $L \neq N$

would require modelling time series either across timespan or about the same timespan but at different discretization rate. To explicitly consider it, we denote an interpolation factor $m_\tau := T_y/T_x$ and an extrapolation factor $m_f := f_y/f_x$ and relate $N$ and $L$ with respect to these factors as follows:

$$\frac{L}{N} = \frac{T_y f_y}{T_x f_x} = m_\tau m_f \tag{1}$$

$m_\tau$ and $m_f$ can be understood as the input-and-output ratio with respect to timespan and discretization rate in modelling, respectively. NFM is designed to effortlessly switching its forward processing between frequency interpolation when $m_\tau > 1$ and frequency extrapolation when $m_f > 1$ with adoption of LFT in **Section 3.2**, which allows an easy specialization of NFM to various time series tasks.

**Preliminary: Discrete Fourier Transform (DFT).** DFT is a computational tool widely used to convert data in physical domain (e.g., time and spatial) to spectral representations. Given the finite-length sequence $x[n \in I_N]$, the DFT and its inverse (IDFT) for recovery to the original sequence, $\mathcal{F}(\cdot)$ and $\mathcal{F}^{-1}(\cdot)$, are defined as follows:

$$X[k] = \mathcal{F}(x) := \sum_{n=0}^{N-1} x[n] e^{-j2\pi kn/N} \tag{2}$$

$$x[n] = \mathcal{F}^{-1}(X) := \frac{1}{N} \sum_{k=0}^{N-1} X[k] e^{j2\pi kn/N} \tag{3}$$

where $k \in I_N$, and the capital letter $X$ is a complex spectral representation of its time-domain variable $x$. Importantly, each $k$ frequency component represents the entirety of the sequence $x$ summarized at different oscillation. The convolution theorem, in conjunction with the IDFT, leverages this characteristic of the FT and provides an efficient way for performing convolution operations through point-wise multiplication in Fourier domain (Rabiner & Gold, 1975; McGillem & Cooper, 1991). We adopt this insight in NFM and introduce a new type of neural Fourier filters (NFFs) playing as a continuous global token mixer that is both expressive and adaptive yet lightweight in **Section 3.3**. Besides, for computation, we utilize an efficient algorithm of DFT, fast Fourier transform (FFT) that offers $\mathcal{O}(N \log N)$ complexity at optimum, as well as its conjugate symmetry property (i.e., $X[N-k] = X^*[k \in I_{K_N}]$). $K_N := \lfloor N/2 \rfloor + 1$ is the number of the first half frequency components of sequence length $N$.

### 3.1 OVERVIEW OF NFM

**Linear frequency interpolation and extrapolation.** We begin with rewriting IDFT in Eq.(3) for the desired output of a model, $z$, as follows:

$$z[n \in I_L] = \frac{1}{L} \sum_{k=0}^{L-1} Z[k] e^{i2\pi kn/L} \tag{4}$$

One straightforward way to express Eq.(4) with respect to the given input sequence $x[n \in I_N]$ would be by taking a linear system to the spectral representation such that $Z[k] = m_\tau m_f (\boldsymbol{W} \mathcal{F}(x[n]) + b)$, where $\boldsymbol{W} \in \mathbb{C}^{K_L \times K_N}$ and $b \in \mathbb{C}^{K_L}$ are weights and bias term, respectively. Directly designing $y = z = \mathcal{F}^{-1}(m_\tau m_f (\boldsymbol{W} \mathcal{F}(x) + b))$ with adoption of a heuristic low-pass filter to further reduce the dimensionality of $\boldsymbol{W}$ and $b$ yields exactly FITS. FITS greatly appreciates the simplicity and lightweight-ness of the linear system and presents a solution to low-resources tasks like edge computing. Nevertheless, it lacks in several aspects as a general solution to a range of time series analysis (refer to **Appendix A**).

**NFM.** We aim to build a general-purpose time series model as a composition $y = \mathcal{P} \circ \mathcal{M}(x)$, that learns mapping between infinite-dimensional spaces of input signal and target function directly in the Fourier domain, given discrete observations $D$. Especially, we introduce NFM for encoder $\mathcal{M}: \mathbb{R}^{N \times c} \to \mathbb{R}^{L \times d}$ that acts globally on the input signal $x$ and seeks the mapping as a frequency interpolation/extrapolation to $z = \mathcal{M}(x)$. The target $y = \mathcal{P}(z)$ is evaluated from the latent representation $z$ through a predictor (e.g., a local-to-local transformation $\mathcal{P}: \mathbb{R}^d \to \mathbb{R}^{d_y}$ and a global-to-local

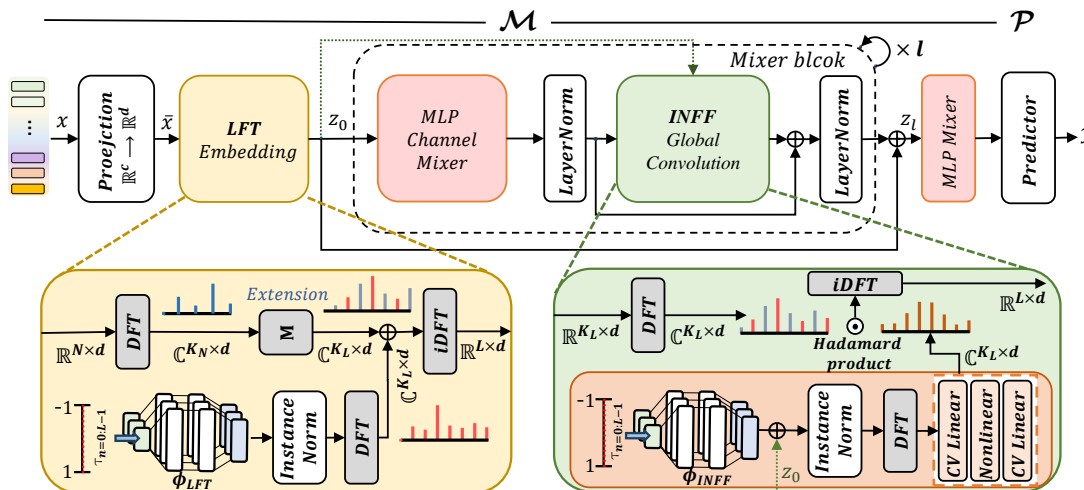

Figure 3: Illustration of NFM architecture consisting of three main learning modules: 1) LFT block to allow flexible frequency extension and provides effective spectral priors, 2) a plain MLP for channel mixing, and 3) INFF module for effective token mixing with global convolution operation. The M in LFT block denotes frequency extension operation.

$\mathbb{R}^{L \times d} \to \mathbb{R}^{d_y}$). As depicted in **Figure 2** and **Figure 3**, the overall workflow of NFM is described in two steps. Given input sequences projected to hidden dimension $\bar{x} \in \mathbb{R}^{N \times d}$ (see **Appendix D**), (i) it is first tailored to temporal embedding $z_0 \in \mathbb{R}^{L \times d} = \mathcal{F}^{-1}(Z_0) = \text{LFT}(\bar{x})$, in that its spectral embedding $Z_0 \in \mathbb{C}^{K_L \times d}$ explicitly accounts for an extension of the original sequence with respect to $m_\tau$ and $m_f$. (ii) Then, the low-level embedding tokens $z_0$ are polished iteratively through a stack of $l$ mixing blocks consisting of a channel-mixing module and global convolution, INFF, as in $z_i = Mixer(z_{i-1})$ for $i = \{1, ..., l\}$. In the following sections, we detail out the two main modules, LFT and INFF.

## 3.2 LEARNABLE FREQUENCY TOKENS

In our framework, we decouple the process of extending the spectral representations of original sequence to that of target domain (i.e., $N \to L$ and $K_N \to K_L$), from learning the abstract coefficients of Fourier interpolation/extrapolation. We denote such extended spectral representation with absence of the weighting coefficients $\bar{Z}_0 \in \mathbb{C}^{K_L \times d}$, and it is obtained by simply initializing the $\bar{Z}_0$ with zeros and rearranging $\bar{X} = \mathcal{F}(\bar{x})$ onto it with scaling for the extension such that $\bar{Z}_0[\lfloor m_t k \rfloor] = m_\tau m_f \bar{X}[k]$ for all $k \in I_{K_N}$, where $\lfloor \cdot \rfloor$ is floor operation.

The result from the above extension is equivalent to the ones from applying zero-padding and/or zero-interleaving to $\bar{X}$. The operation itself is non-parametric and allows handling variable-length input sequence. However, directly adopting $\bar{Z}_0$ is not effective for learning since extending spectral representations with zero coefficients itself does not bring in any information gain. One natural solution to this is introducing extra embeddings that are learned to encapsulate certain abstraction and priors in data during optimization. Indeed, it has become a canonical practice, especially in many Transformer models (Devlin et al., 2018; Chen et al., 2022b; 2023; Wang & Chen, 2020), to enrich the models' learning capability and performance.

Inspired by this, we introduce LFT that can be learned without a-priori and applied directly in the complex Fourier domain as expressive spectral priors across sequences. Especially, we design the LFT by representing the desired spectral priors as a composition of $\mathcal{F}$ and an implicit neural representation (INR) (Sitzmann et al., 2019b; Chen et al., 2021) $\phi: \mathbb{R} \to \mathbb{R}^d$ that maps a temporal location $\tau \in [0, T_y)$ to abstract temporal priors $v$ corresponding to that time location. Notably, the LFT can be characterized as samples from continuous-time Fourier transform (CTFT) of the temporal priors, allowing sampling of the frequency tokens within bandwidth from any arbitrary temporal locations without a need of re-training. We first obtain the learnable frequency tokens $V[k] \in \mathbb{C}^d$ by sampling $v[n] = \phi(\tau_n)$ at temporal locations $\tau_n = \{n/f_y | n \in I_L\}$ and add it to $\bar{Z}_0$ to get $Z_0$. This

entire process of the LFT block is completed as follows:

$$V[k \in I_{K_L}] = \mathcal{F}(\text{InstanceNorm}(\phi(\tau_n))),$$
$$Z_0[k] = (\bar{Z}_0[k] + V[k]), \tag{5}$$
$$z_0[n] = \mathcal{F}^{-1}(Z_0[k])$$

We apply an instance normalization before applying DFT to $v[n]$ to the LFT block as shown in **Figure 3**. This removes DC priors of each channel and a source of internal covariate shift, thereby helping the LFT effectively learn the priors over spectrum - we find in the experiment that the energy of the spectral priors without it is often largely concentrated in DC component, preventing effective learning. The $\phi$ is expressed by MLP with a periodic activation function (Sitzmann et al., 2020) - see **Appendix D**.

### 3.3 IMPLICIT NEURAL FOURIER FILTER

**Neural Fourier filters.** Upon convolution theorem and FT, a convolution kernel can be directly defined in the Fourier domain with arbitrary resolution and its size being as large as the length of inputs. This, then, gives rise to a way for instantiating an efficient global convolution operator $\mathcal{K}\colon (\mathcal{D}, \mathbb{R}^d) \to (\mathcal{D}, \mathbb{R}^d)$ as follows (Guibas et al., 2021; Li et al., 2020):

$$\mathcal{K}(z)(\tau) = \int_{\mathcal{D}} \kappa(\tau - s)z(s)\, \mathrm{d}s, \quad \forall \tau \in \mathcal{D} \tag{6}$$

$$\mathcal{K}(z)(\tau) = \mathcal{F}^{-1}(\mathcal{R} \cdot \mathcal{F}(z))(\tau), \quad \forall \tau \in \mathcal{D} \tag{7}$$

where Eq.(6) and Eq.(7) express convolution operator on physical space of the signals and its equivalent form with convolution theorem and the FT applied, respectively. $\mathcal{R} \equiv \mathcal{F}(\kappa)$ is a Fourier filter defined directly in the Fourier domain and parameterized by a neural network. While a shallow fully-connected network is generally sufficient to parameterize $\mathcal{R}$, much design concerns are put in how to achieve properties of NFFs that are desirable for modelling. We summarize them and compare the existing NFFs from the designing perspective in **Appendix B**. In short, it is highly desirable to have a NFF that is memory-efficient, length-independent (i.e., flexible), instance-adaptive, and mode-adaptive (i.e., expressive and generalized across spectrum). We design INFF that satisfy these properties below.

**INFF.** INFF is formulated for modulating the embedding tokens $z$ globally in the Fourier domain to $\hat{z}$ through a Fourier filter, $\mathcal{R}[k] \in \mathbb{C}^d$, as follows:

$$\hat{z} = \mathcal{F}^{-1}(\mathcal{R}(z_0) \odot \mathcal{F}(z)) \tag{8}$$

where $\odot$ denotes Hadamard product. The computation of $\mathcal{R}$ is conditioned on the initial spectral embedding $Z_0$, for which a reason will be clarified later. Here, the use of an INR for encapsulating abstract spectral priors in **Section 3.2** is extended to define $\mathcal{R}$ with implicit filter coefficients.

$$\mathcal{R}(z_0) \coloneqq \mathcal{W}(\mathcal{F}(\text{InstanceNorm}(\phi(\tau_n) + z_0))) \tag{9}$$

Recalling that $\mathcal{F}(\phi(\tau_n))$ implicitly models CTFT, in this formulation the Fourier filter has filter coefficients defined uniquely for each period in spectrum, but with a *single parameterization*. That is, the designed Fourier filter is in a compact form and can readily handle variable-length sequence with unique frequency coefficients (i.e., mode-adaptive). Unfortunately, the INFF with the filter solely defined by the $\mathcal{F}(\phi(\tau_n))$ would lack expressivity due to the reliance on depth-wise convolution (i.e., no channel mixing) and struggle to generalize across different instances. We further improve the filter on both factors by aggregating its temporal coefficients with the initial temporal embedding tokens $z_0$ and processing them through a complex-valued MLP, $\mathcal{W}\colon \mathbb{C}^d \to \mathbb{C}^d$. Note that we use ReLU for non-linearity and have no bottleneck or expansion factor for the intermediate dimension. With this, each feature of the updated $k$th filter coefficient now represents a mixture of all features of $k$th implicit filter coefficient and all features of the $k$th spectral representation of the input embedding.

Finally, INFF is put together with a plain channel-mixing block and Layer Normalization to form a complete Mixer block. We configure the components as pre-channel-mixing and post-normalization with a skip connection in each Mixer block and stack multiple Mixer blocks followed by a final channel-mixing block to constitute a NFM backbone network $\mathcal{M}(\cdot)$ as shown in **Figure 3**.

Table 1: Long-term forecasting results averaged over 4 horizons. The best averaged results are in **bold** and the second best are underlined. Full result is available in **Appendix E**.

| Model (params) | NFM (27K) | | FITS (∼0.2M) | | N-Linear (∼0.5M) | | iTransformer (∼5.3M) | | PatchTST (∼8.7M) | | TimesNet (∼0.3B) | |
|---|---|---|---|---|---|---|---|---|---|---|---|---|
| Metric | MSE | MAE | MSE | MAE | MSE | MAE | MSE | MAE | MSE | MAE | MSE | MAE |
| ETTm1 | **0.345** | **0.375** | 0.357 | 0.380 | 0.366 | 0.383 | 0.371 | 0.401 | 0.353 | 0.382 | 0.400 | 0.406 |
| ETTm2 | **0.250** | **0.311** | **0.250** | 0.313 | 0.258 | 0.315 | 0.276 | 0.337 | 0.256 | 0.317 | 0.291 | 0.333 |
| ETTh1 | **0.407** | **0.420** | **0.407** | **0.420** | 0.413 | 0.422 | 0.503 | 0.491 | 0.413 | 0.434 | 0.458 | 0.450 |
| ETTh2 | 0.356 | 0.400 | 0.334 | 0.382 | 0.343 | 0.389 | 0.405 | 0.430 | **0.331** | **0.381** | 0.414 | 0.427 |
| Weather | **0.227** | 0.269 | 0.241 | 0.280 | 0.254 | 0.288 | 0.255 | 0.289 | **0.227** | **0.264** | 0.259 | 0.287 |
| Electricity | **0.159** | **0.251** | 0.163 | 0.254 | 0.169 | 0.262 | 0.163 | 0.259 | **0.159** | 0.253 | 0.193 | 0.298 |
| Traffic | 0.391 | **0.260** | 0.411 | 0.280 | 0.433 | 0.290 | **0.376** | 0.270 | 0.391 | 0.264 | 0.620 | 0.336 |

## 4 EXPERIMENTS

We conduct extensive experiments to demonstrate the effectiveness of NFM and its competence as a general solution to time series modelling. Note that we provide only a brief description about the setting for each experiment below, but one can find all details in the supplementary section (**Appendix C∼E**). Our code is publicly available at: https://github.com/minkiml/NFM.

**Implementation.** A general-purpose time-series model allows one to address a range of tasks as well as various time series modalities without significant architectural modifications (i.e., no injection of task-specific inductive bias). To this end, we implement a single NFM backbone $\mathcal{M}(\cdot)$ and use it for all tasks (only differ by some hyper-parameters such as hidden size and the number of mixer blocks) by equipping it with a task-specific linear predictor $\mathcal{P}(\cdot)$ (simply a fully-connected layer). For all experiments, we use a single NVIDIA A100 GPU.

### 4.1 TIME SERIES MODELLING

We begin with showcasing the efficacy of NFM in three distinct time-series tasks, including long-term forecasting, anomaly detection, and classification under their conventional scenario ($f_x^{train} = f_x^{test}$). We follow the experimental setups: forecasting (Zhou et al., 2021), anomaly detection (Xu et al., 2021a), and classification (Romero et al., 2021). Refer to **Appendix D** for details.

**Long-term forecasting.** Long-term times-series forecasting task ($m_f = 1$ and $m_\tau > 1$) is conducted on 7 benchmark datasets over 4 horizons (96, 192, 336, and 720). In **Table 1**, The averaged results of NFM on MSE and MAE metrics are compared with 5 SOTA forecasting models - FITS (Xu et al., 2023), N-Linear (Zeng et al., 2023), iTransformer (Liu et al., 2023), PatchTST (Nie et al., 2022), TimesNet (Wu et al., 2022). While the performance of NFM is highly competitive with that of the dedicated forecasting models with hundreds or millions of trainable parameters, we put an extra emphasis on its compactness. Especially, the number of parameters that need to be trained in NFM is only around **27K** for all forecasting cases, which sheds light on both low-resource on-device learning and processing. This result stands out that of the simple linear models like N-Linear and FITS, and other SOTA by large margin, for which the number of parameters is subject to the length of both lookback and prediction horizon whereas the number of parameters in NFM is decoupled from the length of input or target prediction. Please see **Appendix E** for more analysis about the results.

**Anomaly detection.** We frame the anomaly detection task as learning correct contexts (dominant and normal contexts) and evaluating the validity of observations within the contexts. More specifically, our approach is to learn the correct contexts by reconstructing complete sequences from their down-sampled counterparts (i.e., $m_f > 1$ and $m_\tau = 1$) - refer to **Appendix D** for detail. For quantitative evaluation, we employ 4 popular anomaly detection benchmark datasets and compare the performance with 6 baselines - vanilla Transformer (Vaswani et al., 2017), PatchTST (Nie et al., 2022), TimesNet (Wu et al., 2022), ADformer (Xu et al., 2021a), N-linear (Zeng et al., 2023), and FITS (Xu et al., 2023). In **Table 2**, NFM shows effectiveness in anomaly detection task, presenting top-tier results in three datasets (SMD, MSL, and PSM) and third-tier result in SMAP. This result in NFM is achieved with a compact model size of only around **6.6K** parameters, which follows that of FITS (1.3K) and stands out that of the deep feature learning models (TimesNet and ADformer) by considerably large

Table 2: Anomaly detection results (F1-score) on 4 datasets, where higher F1-score indicates better performance. Full tabular result is available in **Appendix E**.

| Model (params) | SMD | MSL | SMAP | PSM |
|---|---|---|---|---|
| Transformer (0.2M) | 75.95 | 81.93 | 69.70 | 88.75 |
| PatchTST (0.2M) | 82.11 | 80.51 | 69.11 | 96.00 |
| TimesNet ($\sim$28M) | 83.27 | 81.70 | **73.23** | 97.30 |
| ADformer*(4.8M) | 76.38 | 81.78 | 71.18 | 83.14 |
| N-Linear (10K) | 81.81 | 81.18 | 67.50 | 95.77 |
| FITS (1.3K) | 81.67 | 80.77 | 64.07 | 96.60 |
| **NFM** (6.6K) | **84.32** | **82.46** | 70.88 | **97.51** |

*The joint criterion in ADformer is replaced with the simple reconstruction error to compute anomaly score for fair comparison.

Table 3: Classification accuracy (%) on Speech-Command. $\sim$ denotes inapplicable (prohibitively slow) or computationally not possible on single GPU.

| Model (params) | SpeechCommand | | |
|---|---|---|---|
| | MFCC | RAW (SR=1) | RAW (SR=1/2) |
| ODE-RNN (89K) | 65.90 | $\sim$ | $\sim$ |
| GRU-$\Delta t$ (89K) | 20.0 | $\sim$ | $\sim$ |
| GRU-ODE (89K) | 44.8 | $\sim$ | $\sim$ |
| NCDE (89K) | 88.5 | $\sim$ | $\sim$ |
| NRDE (89K) | 89.8 | 16.49 | 15.12 |
| S4 (400K) | 93.96 | **96.17** | **94.11** |
| CKConv (100K) | **95.27** | 71.66 | 65.96 |
| Transformer (800K) | 90.75 | $\sim$ | $\sim$ |
| **NFM** (37K) | 94.23 | 90.94 | 90.30 |

Table 4: Forecasting results (MSE) and performance drops (%) at different sampling rate. Three models, including PatchTST (Transformer-based), N-Linear (time-domain linear model), and FITS (frequency-domain linear model) are opted for comparison. The best performance is in blue and the least performance drop in red. Full tabular results over all horizons can be found in **Appendix E**.

| Model | SR | ETTm1 | | ETTm2 | | Weather | |
|---|---|---|---|---|---|---|---|
| | | 96 | 720 | 96 | 720 | 96 | 720 |
| PatchTST | 1/4 | 0.394 (34.5 ↓) | 0.433 (4.1 ↓) | 0.232 (39.8 ↓) | 0.382 (5.5 ↓) | 0.212 (42.3 ↓) | 0.329 (4.8 ↓) |
| | 1/6 | 0.434 (48.5 ↓) | 0.443 (6.5 ↓) | 0.265 (59.6 ↓) | 0.396 (9.4 ↓) | 0.236 (58.4 ↓) | 0.335 (6.7 ↓) |
| N-Linear | 1/4 | 0.436 (42.5 ↓) | 0.469 (8.3 ↓) | 0.288 (72.5 ↓) | 0.409 (11.1 ↓) | 0.268 (47.3 ↓) | 0.351 (3.8 ↓) |
| | 1/6 | 0.482 (57.5 ↓) | 0.471 (8.8 ↓) | 0.281 (68.3 ↓) | 0.422 (14.7 ↓) | 0.280 (53.8 ↓) | 0.364 (7.7 ↓) |
| FITS | 1/4 | 0.348 (12.6 ↓) | 0.426 (2.9 ↓) | 0.198 (21.5 ↓) | 0.356 (2.0 ↓) | 0.182 (7.7 ↓) | 0.325 (1.2 ↓) |
| | 1/6 | 0.368 (19.1 ↓) | 0.437 (5.3 ↓) | 0.216 (31.7 ↓) | 0.364 (4.3 ↓) | 0.195 (14.8 ↓) | 0.329 (2.5 ↓) |
| **NFM** | 1/4 | 0.299 (4.5 ↓) | 0.407 (0.2 ↓) | 0.179 (11.9 ↓) | 0.356 (2.0 ↓) | 0.164 (6.5 ↓) | 0.315 (1.0 ↓) |
| | 1/6 | 0.319 (11.5 ↓) | 0.414 (2.0 ↓) | 0.189 (18.2 ↓) | 0.358 (2.6 ↓) | 0.173 (12.3 ↓) | 0.318 (1.9 ↓) |

margin. Note that the remarkably small model size of FITS is highly subject to cases (short length of input and target while their energy dominantly spans in low frequency regime).

**Classification.** We evaluate the effectiveness of NFM in time-series classification task ($m_f = 1$ and $m_\tau = 1$) using *SpeechCommand* dataset (Warden, 2018) that provides both MFCC features ($N = 161$) and raw waveform ($N = 16$k). In **Table 3**, we report the classification accuracy (ACC) and compare it with that of 8 different baselines (7 continuous-time models and 1 Transformer) - ODE-RNN (Rubanova et al., 2019), GRU-$\Delta t$ (Kidger et al., 2020), GRU-ODE (De Brouwer et al., 2019), NCDE (Kidger et al., 2020), NRDE (Morrill et al., 2021)), S4 (Gu et al., 2021), CKConv (Romero et al., 2021), and vanilla Transformer (Vaswani et al., 2017). Overall, NFM yields competitive performance in both cases of processing MFCC features - CKConv (95.27%) vs. NFM (**94.23%**) vs. S4 (93.96%), and raw waveform - S4 (96.17%) vs. NFM (**90.94%**) vs. CKConv (71.66%) with far smaller model size (**37K**) among all baselines. The neural ODE-based models are generally in a compact form with a weight-tying architecture but struggle dealing with long series due to the need of solving differential equations for a long step. Vanilla transformer suffers from huge memory occupancy and thus is unable to process the raw waveform in a single GPU setting.

## 4.2 EVALUATION AT DIFFERENT DISCRETIZATION RATES

In practice, it is highly desirable for a model to have a resolution-invariance property that readily enables generalization of the learned solutions to unseen discretizations without significant performance lose. To this end, we now reveal this aspect of NFM, which learns function-to-function mappings, by conducting the classification and forecasting tasks in a scenario where the observations are sampled at different (unseen) sampling rate during testing time (SR= $f_x^{test}/f_x^{train}$). The classification result in **Table 3** shows that NFM has the least performance drop, yielding only around **0.7% ↓** degradation

from the original performance on the input sequences sampled at unseen sampling rate of SR= $1/2$, compared to NRDE (**8.31%** ↓), CKConv (**7.95%** ↓), and S4 (**2.14%** ↓). In the forecasting task, the similar result is drawn as shown in **Table 4**, where the performance degradation of NFM in all cases is considerably lower than that of the 3 baselines. We provide more details on the forecasting results in **Appedix E**.

## 4.3 EXPLORATION OF NFM

**Effects of LFT and INFF.** We examine the effect of the proposed LFT in NFM (*INFF + LFT*) by comparing a NFM with randomly initialized complex-valued learnable weights as frequency tokens (namely, *Naive*) and without frequency tokens (namely, *INFF-only*). As shown in **Figure 4**, the improvement on performance is remarkable, reporting **82.6** → **90.9** (**10.0%** ↑) and **79.4** → **90.9** (**14.5%** ↑) compared to *Naive* and *INFF-only*, respectively. In the case of testing at different input resolution (SR=1/2), the improvement is more significant, yielding **78.6** → **90.4** (**15.0%** ↑) and **74.1** → **90.4** (**22.0%** ↑) respectively. These results from *INFF + LFT* are achieved with ∼ 8.6 times smaller and ∼ 1.16 times larger model size compared to the *Naive* and *INFF-only*, which shows

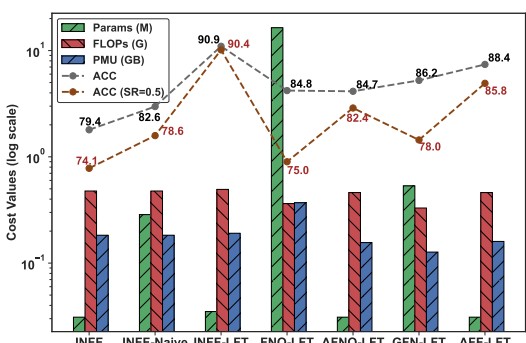

Figure 4: Comparison of NFM with different ablation cases on SpeechCommand dataset. PMU is peak memory usage during inference time.

that LFT is conducive to learning spectral priors in a compact form. Furthermore, we compare INFF with 4 different SOTA NFFs, including FNO (Li et al., 2020), AFNO (Guibas et al., 2021), GFN (Rao et al., 2021), AFF (Huang et al., 2023). The result in **Figure 5** shows the overall performance of *INFF + LFT* surpasses the *others + LFT* by notable margin. The instance-adaptive NFFs - AFNO (**2.7%** ↓) and AFF (**2.9%** ↓), perform more robustly against different input resolution than the mode-adaptive ones - FNO (**11.6%** ↓) and GFN (**9.5%** ↓). As analysed in **Appendix B**, INFF which is both instance- and mode-adaptive further improves the robustness (**0.7%** ↓). This superiority of INFF comes only at the cost of a minor complexity and memory usage increase. Besides, it is noteworthy that dealing with different input resolution is originally not possible in the mode-adaptive ones (FNO and GFN) due to the fixed-length operation without a heuristic low-pass filter, but becomes possible with the adoption of LFT in NFM framework. The similar results are obtained on forecasting task (see **Appendix E** for it and full tabular results).

**Visualization on INFF.** We study the behaviour of INFF in terms of what representations it potentially leads to be learned. Specifically, one would expect to see INFF learning effective filter coefficients such that INFF amplifies frequencies of relevant information while suppressing frequencies of irrelevant one. To confirm this, we synthesize simple single channel band-limited (up to Nyquist frequency $f_{nyquist}$) signals by composing multiple frequency components. During generation, we assign class labels to them according to certain combinations of the frequency components spanning in a frequency range $[f_A, f_B(< f_{nyquist})]$. Please refer to **Appendix C** for details about the generation. For experiment, we sample 10 classes of sequences of length $N = 2000$ at $f_x = N$ with the class frequencies spanning in the range $[320, 590]$. **Figure 5** provides a visualization of the resulting INFF trained on the synthetic data, where it is clearly shown that finding a correct

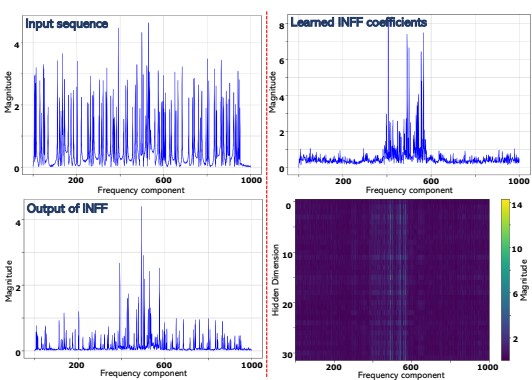

Figure 5: Visualization of INFF on synthetic data. The **top-left** figure shows the frequencies of input sequence, **bottom-left** the frequencies of filtered sequence, **bottom-right** the learned INFF's coefficients, and **top-right** the coefficients averaged over hidden dimension.

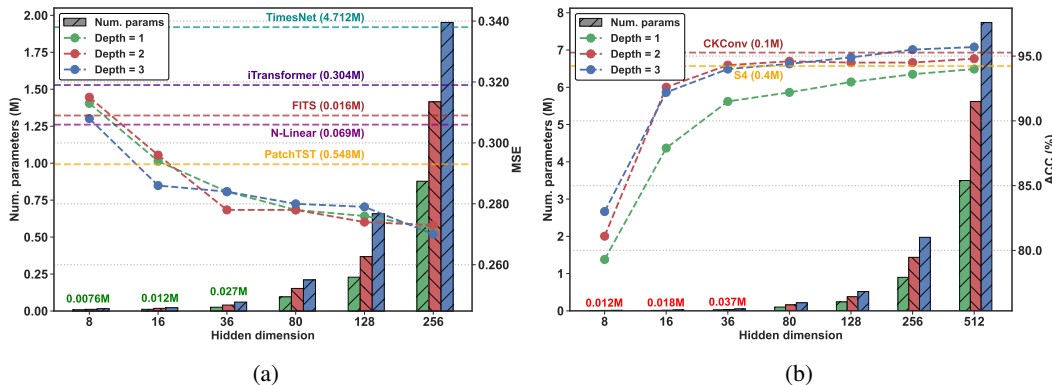

(a)                                    (b)

Figure 6: Scaling behavior (performance - the line plots) of NFM with respect to varying hidden dimension and depth on (a) ETTm1 (forecasting over the horizon of 96) and (b) SC-MFCC (classification) datasets. The bar plots represent the number of parameters computed at each set of hidden dimension and depth. Baseline performances are also included with the dashed horizontal lines for comparison.

solution comes with INFF learning its filter coefficients aligned with the frequency range $[320, 590]$ of the class labels (relevant information).

**Scaling behaviour and compactness of NFM.**   We present how scaling over hidden dimension ($d$) and depth affects NFM's performance in **Figure 6**. While it is clearly seen that increasing the NFM's scale consistently leads to improved performance, we observe that NFM rapidly reaches a regime of competitive performance despite having significantly fewer parameters compared to all baseline models (as indicated by the dashed horizontal lines). This highlights the compactness of NFM, which achieves high performance without the need for excessive parameter scaling.

## 5   CONCLUSION

In this work, we have introduced, NFM, modelling time series directly in Fourier domain by formulating the Fourier-domain data manipulation into Fourier interpolation and extrapolation. NFM with *learnable frequency tokens* and *implicit neural Fourier filter* enjoys the intriguing properties of the FT in learning, resulting in a remarkable compactness and continuous-time characteristics. Our experiments demonstrate that NFM can be a powerful general solution to time series analysis across a range of datasets, tasks, and scenarios and show that it is possible to achieve the state-of-the-art performance with a remarkably compact model, compared to models with hundreds thousand and million parameters.

**Limitations and Future Work.**   Despite the fact that NFM models time series in function space through Fourier lens, the current implementation is not directly suitable for some other dynamic scenarios such as handling irregular time series. A reason for this is that the FFT for efficient transformation requires uniformly-sampled sequence thus not applicable while a naive algorithm with computing a DFT matrix and its pseudo inverse is not only too slow but also memory-intensive for every long and multivariate but irregular time series. While addressing this challenging scenario is highly valuable in time series analysis, we are currently improving NFM on this matter.

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

## A   APPENDIX: COMPARISON WITH FITS

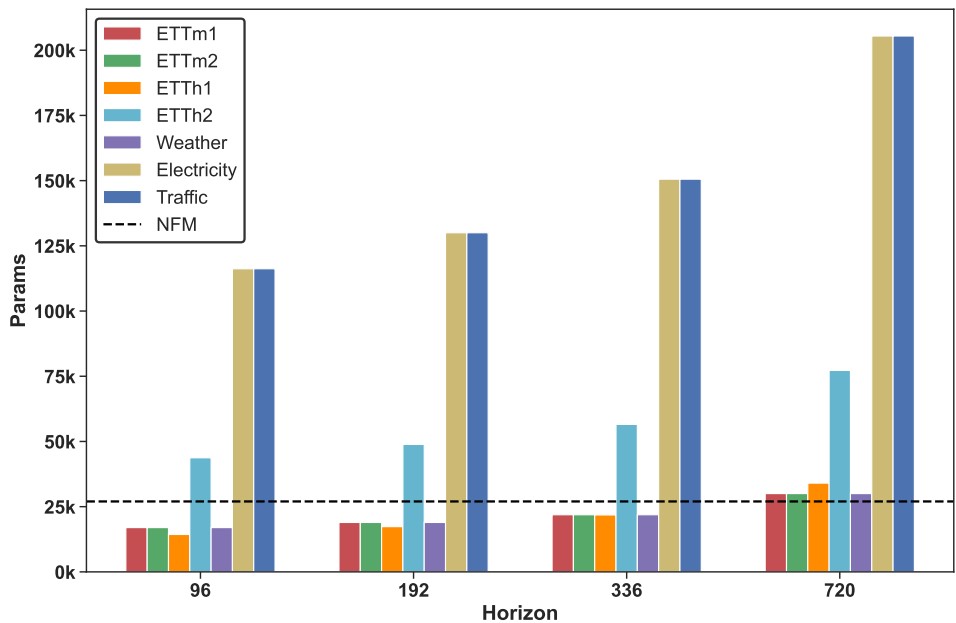

Figure 7: The number of parameters in NFM (**dashed line**: 27K) and FITS (**colour bars**) for different length of the prediction horizons (lookback of 720 for all cases except for ETTh1 for which the lookback window is 360). The number of parameters in FITS is to yield the results in the main performance table.

We provide an additional discussion on a recent time series model, Frequency Interpolation Time Series analysis baseline (FITS), that is introduced on the same principle of frequency-domain manipulation as NFM.

FITS is designed with a single complex-valued linear layer to directly learn the linear interpolation or extrapolation from input frequency spectrum as analysed in **Section 3.1**. It is equipped with a heuristic low-pass filter which restricts the spectrum and further reduces the model complexity. While FITS achieves an elevated degree of lightweight-ness with competing performance in some time series tasks, there present several limitations to be improved for broader utilization.

- FITS cannot model multivariate time series. With its simplified architecture, it relies on channel-independent modelling (Nie et al., 2022) which does not model correlation between channels, thus its application is limited to univariate scenarios only.

- FITS is a linear model operating directly in frequency domain, and thus its expressiveness and learning capacity are largely limited. This is especially disadvantageous when it comes to modelling large-scale dataset and more complex patterns. Moreover, the complexity of FITS itself without a low-pass filter increases exponentially with the length of inputs and target outputs (there is a trade-off between giving up some information and resolving the complexity with the use of a low-pass filter). This characteristic does not allow FITS to stay compact in many practical scenarios with more complex and/or long time series whose major frequencies spread in wide spectrum. This is well shown from the foreacsting results on electricity and traffic datasets (**Table 7**). In **Figure 6**, we also provide the overall number of parameters in NFM and FITS used in forecasting task. Taking any lower cut-off frequency for the low-pass filters in the datasets like ETTh2, elctricity, and traffic to make FITS more compact leads to considerable performance drops.

- As discussed in the main body of this work, one natural advantage gained from modelling time series directly in the Fourier domain is that the learned function can be more robust to dealing with change in discretization of input data. These features, however, are difficult to

be fully exploited in FITS due to the static nature of the model (not able to deal with variable length of time series). Although the adoption of a low-pass filter mitigates this issue by allowing to handle any length of input sequence longer than the period of cut-off, defaulting a model with a heuristic low-pass filter is not always practical. It discards the information of all higher frequency components, and thus costs performance. Indeed, in our experiment, the performance with full spectrum is consistently better than that with restricted spectrum.

To address the aforementioned limitations in FITS, entirely renovating the model would be necessary. In one sense, NFM is a complete renovation and generalization of FITS with a compact, lightweight, and adaptive deep feature learning module. It is not only designed to model both multivariate and univariate time series but scalable to learn more complex patterns in data. Notably, NFM can be implemented in a very compact form (less than 40K for all tasks in the experiments) regardless of the input and target output length and performs consistently well in various time series tasks. This feature of NFM stands out the compactness of FITS and other linear models like (Zeng et al., 2023) - the compactness with respect to performance of FITS would only be better than NFM in some unique cases where the processed input length is sufficiently short and/or an effective heuristic low-pass filter with low cut-off frequency can be chosen.

## B   APPENDIX: ANALYSIS OF NEURAL FOURIER FILTER DESIGNS

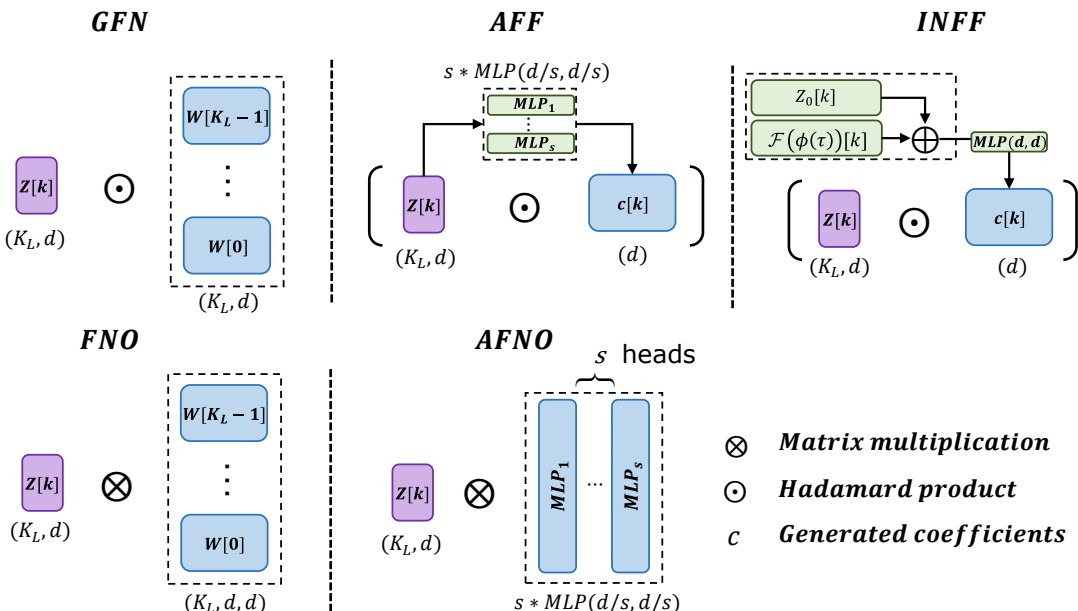

Figure 8: Overview of different NFF designs.

NFFs can mainly be found in the context of operator learning to solve PDEs and global token mixing as an efficient alternative to attention mechanism in vision tasks. Here, we summarize the existing NFFs from design perspective as shown in **Figure 7** and compare them with NFM.

**FNO** (Li et al., 2020) is designed with per-mode matrix multiplication. Such per-mode parameterization allows the model to be expressive with large learning capacity. However, the model can also be easily over-parameterized and thus the advantage comes with a high risk of overfittig. Moreover, the inhibitive number of parameters can be incurred when the model needs to deal with long sequences. Besides, the per-mode parameterization essentially prevents the model from being adaptive to handling variable-size inputs unless a heuristic low-pass filter is integrated.

**GFN** (Rao et al., 2021) also adopts per-mode multiplication but of entry-wise operation (i.e., depth-wise global convolution). While this features GFN with more efficient parameterization than FNO it

raises another concern of landing no channel-mixing operation in learning and may limit the model's expressivity. Additionally, due to the per-mode parameterization, GFN is limited to a scenario where the input length is static as well, hence it is less suitable for time-series tasks. Note that FEDformer (Zhou et al., 2022b) and FourierGNN (Yi et al., 2024a) are good examples of GFN adapted to time series problems with random frequency masking and just a single set of filter coefficients, respectively, which resolve the issue with the static input length but would still suffer from the lack of expressivity.

**AFNO** (Guibas et al., 2021) handles variable-size inputs and achieves channel mixing simply by having a single parameterization (1×1 convolution + non-linearity + soft-shrinkage function) and sharing its weights across all frequency modes in spectrum. AFNO can essentially be seen as a generalization of FNO and GFN with a block-diagonal (i.e., multi-head) structure, Note that FreTS (Yi et al., 2024b) adopts AFNO. However, it trades off expressivity (i.e., there is no principled way for AFNO with a single shared network to learn discriminative feature across frequency modes) over efficiency and flexibility to handling variable-size inputs. Besides, a soft shrinkage function is adopted in AFNO as a regularization to encourage sparsity, with which we, however, consistently observe a performance drop on time series tasks in all NFFs including INFF as well.

The weights of the filter (i.e., filter coefficients) in the above works are all shared across different instances. Unfortunately, this poses a concern that the models can struggle to learn complex patterns across different instances that reside upon different underlying spectrum.

**AFF** (Huang et al., 2023) addresses this by altering the use of neural network from directly parameterized filter coefficients to a hypernetwork (Ha et al., 2017) that yields the filter coefficients dynamically according to each input instance.

Based on the above analysis, one remaining gap in the current NFFs we found is how to design a filter that is aware of each mode separately in spectrum, namely "mode-adaptive", and modulate them locally in efficient way. As discussed, FNO and GFN achieve such mode-adaptivity by instantiating the filter coefficients separately for each mode. However, this way not only makes the operation static (due to fixed parameterization) but also come at the cost of significantly increasing the number of parameters (especially, when operating matrix multiplication as in FNO) with the input length. On the other hand, AFNO and AFF do not account for the mode-adaptivity while simply sharing filter coefficients for all modes.

Our proposed INFF essentially gives a solution to this matter without sacrificing other favourable NFFs' characteristics. We achieve this by leveraging an INR (Sitzmann et al., 2019a; Tancik et al., 2020b; Sitzmann et al., 2020), a neural-based technique to reformulate a direct representation of interests into a more compact form of implicit representation. Especially, in INFF, the per-mode parameterization of a Fourier filter is turned into a process of learning to encode an abstract and implicit representation of the whole filter coefficients in spectrum. Thus, it is possible to draw unique filter coefficients for any frequency modes within the bandwidth at the cost of only a single parameterization, which largely contributes to achieving high compactness in NFM.

## C APPENDIX: SYNTHETIC DATA FOR INFF VISUALIZATION

We synthesize simple single-channel band-limited signals $x$ of $K$ classes by composing $M = S + R$ frequency components within its Nyquist frequency, $f_{nyquist} = f_x/2$. For $k$th class, we first create a signal from a set of fixed $S$ frequency components $\{f_1^k, ..., f_S^k\}$ of the class that are randomly chosen from a frequency range $[f_A, f_B(< f_{nyquist})]$. Then, we further combine the signal with different sets of $R$ frequency components $\{f_1, ..., f_R\}$ drawn randomly from $U\{1, ..., f_{nyquist}\}$ to create different variants of the class signal. This generation of $k$th class signal is expressed as follows:

$$x^k(\tau) = \sum_{i=1}^{S} A_i^k sin(2\pi f_i^k \tau + \theta_i^k) + \sum_{j=1}^{R} A_j sin(2\pi f_j \tau + \theta_j) + \delta(\tau) \tag{10}$$

where $A \sim U(0, 1)$ and $\theta$ are amplitude and phase components, respectively, and $\delta \sim N(0, \sigma^2)$ is gaussian noise. For the synthetic data used in experiment, we set $\theta$ to a constant value, $K = 10$, $S = 20$, and $R = 40$ and for each class signal we generate 100 sequence samples of length $N = 2000$ (with $T_x = 1$ and $f_x = N$).

# D APPENDIX: IMPLEMENTATION AND EXPERIMENTAL DETAILS

## D.1 IMPLICIT NEURAL REPRESENTATIONS IN NFM

INR is parameterized by a neural network and trained to represent a target instance as a continuous function that maps grid-based representations (e.g., spatial coordinates or temporal locations), $\tau \in \mathbb{R}^{d_{in}}$, to the corresponding feature representations. A general formulation of INRs, $\phi\colon \mathbb{R}^{d_{in}} \to \mathbb{R}^d$, is based on a $L$ layers MLP and can be expressed as follows (Yüce et al., 2022):

$$
\begin{aligned}
z_{INR}^{(0)} &= \gamma(\tau), \\
z_{INR}^{(l)} &= \alpha^{(l)}(\boldsymbol{W}^{(l)} z_{INR}^{(l-1)} + b^{(l)}), (l = 1, ..., A - 1) \\
\phi(\tau) &= \boldsymbol{W}^{(L)} z_{INR}^{(A-1)} + b^{(A)}
\end{aligned}
\tag{11}
$$

where $\gamma\colon \mathbb{R}^{d_{in}} \to \mathbb{R}^{h_0}$ is an initial feature encoding function, and $\boldsymbol{W}^{(l)} \in \mathbb{R}^{h_l \times h_{l-1}}$, $b^{(l)} \in \mathbb{R}^{h_l}$, and $\alpha^{(l)}$ are weights, bias, and an element-wise non-linear activation, respectively. For the INR layer in LFT and INFF modules of NFM, we opt for SIREN Sitzmann et al. (2020) - a MLP with $\alpha = sin(\cdot)$ and $z_{INR}^{(0)} = sin(w_0(\boldsymbol{W}^{(0)} r + b^{(0)}))$, where $w_0$ is a constant that controls the frequency region of the activation.

Implementing the $\phi(\cdot)$ solely based on SIREN requires putting a care on finding a good $w_0$ to deal with spectral bias - a tendency of MLPs towards prioritizing learning low-frequency components of the features (Rahaman et al., 2019). In order to alleviate this, as in (Kim et al., 2023) we further incorporate Fourier features (Tancik et al., 2020a; Mildenhall et al., 2021) into the $\phi(\cdot)$ by replacing the $\gamma(\tau) = sin(w_0(\boldsymbol{W}^{(0)}\tau + b^{(0)}))$ with $[sin(2\pi a_1\tau), cos(2\pi a_1\tau), ..., sin(2\pi a_{h_0/2}\tau), cos(2\pi a_{h_0/2}\tau)]$ where we sample $\{a_i\}_{i=1}^{h_0/2} \sim \mathcal{N}(0, 128)$.

In the experiments, we find that small $\phi$ works sufficiently well and do not see any noticeable improvements with larger $\phi$ in performance. For all tasks, we implement $\phi$ in LFT and INFF with the same settings - the temporal locations $\tau$ sampled equidistantly from the range $[-1, 1]$, dimension of the input temporal location $d_{in} = 1$, the number of layers $A = 3$, and hidden unit dimension $h_0 \to h_1(32) \to h_2(32) \to h_3(d)$, where $h_0$ varies with datasets.

## D.2 INPUT PROJECTION

Instead of the projecting input temporal features $x$ to initial hidden embeddings $\bar{x}$ solely through a matrix multiplication and passing down to the main processing modules, we combine it with non-linear projection of a periodic activation using the same formulation as SIREN shown in **Appendix D.1**.

$$
\bar{x} = \boldsymbol{W}_l x + \boldsymbol{W}_n^{(2)}(sin(w(\boldsymbol{W}_n^{(1)} x + b_n^{(1)})))
\tag{12}
$$

where $\boldsymbol{W}_l \in \mathbb{R}^{d \times c}$, $\boldsymbol{W}_n^{(1)} \in \mathbb{R}^{* \times c}$, $\boldsymbol{W}_n^{(2)} \in \mathbb{R}^{d \times *}$, $b_n^{(1)} \in \mathbb{R}^{*}$, and $w$ is a frequency scaling factor. From the experiments, we find that relying solely on the projection through $\boldsymbol{W}_l$ was not effective for both channel-independent and multi-channel cases - this could be due to the inherent information sparsity of time series when working out of the features of each temporal location independently (Li et al., 2023; Nie et al., 2022). A natural alleviation as a very common practice to this, especially with Transformer-based models but not limited to, is to employ patchification (e.g., Nie et al. (2022)) before the projection. However, this brings in a concern. While patchification reliefs the information sparsity in the inputs by forcing to reform them with a strong inductive bias of some "locality", the process is essentially sensitive to the change in temporal structure (e.g., resolution and arrangement) of the input time series. Thus, this factor hinders learning continuous-time characteristics in models.

Regarding learning, the projection in Eq.12 can be seen to be enriching each temporal feature (point-wise input token) independently through initial channel mixing at different periods. In the experiments, we see that the performance of NFM improves by a noticeable margin across all datasets and tasks. We use both sine and cosine activation and set $w$ simply to 1 (with higher $w$ the performance tends to degrade in our experiments).

## D.3 GENERAL IMPLEMENTATION AND HYPERPARAMETER CONFIGURATIONS

Table 5: Hyperparameter settings for different experiments. ADs denotes all anomaly detection datasets. Note that the batch size of the forecasting datasets is set large as channel-independence is applied.

| Params | ETTm1&2 | ETTh1&2 | Weather | Electricity | Traffic | SpeechCommand | | ADs |
|---|---|---|---|---|---|---|---|---|
| | | | | | | Raw | MFCC | |
| Epochs | 40 | 40 | 40 | 40 | 40 | 300 | 300 | 150 |
| Batch | 1792 | 896 | 1680 | 1648 | 1648 | 160 | 240 | 128 |
| Optimizer | Adam | Adam | Adam | Adam | Adam | Adam | Adam | Adam |
| Weight Decay | - | - | - | - | - | - | - | - |
| LR scheduler | - | - | - | cosine | cosine | cosine | cosine | - |
| Learning rate ($\times e^{-4}$) | 2.0 | 1.5 | 2.5 | $3.5 \rightarrow 1.5$ | $3.5 \rightarrow 1.5$ | $7.5 \rightarrow 3.5$ | $5.0 \rightarrow 2.5$ | 1.0 |
| Dropout | $0.05 \sim 0.35$ | $0.05 \sim 0.35$ | 0.15 | 0.15 | 0.15 | 0.05 | 0.25 | - |
| Patience | 6 | 6 | 6 | 3 | 3 | 30 | 30 | 10 |
| Mixer blocks | 1 | 1 | 1 | 1 | 1 | 2 | 2 | 1 |
| Hidden size ($d$) | 36 | 36 | 36 | 36 | 36 | 32 | 32 | 8 |
| $h_0$ | 32 | 32 | 32 | 32 | 32 | 32 | 32 | 16 |
| Predictor $\mathcal{P}(\cdot)$ | $d \rightarrow 1$ | $d \rightarrow 1$ | $d \rightarrow 1$ | $d \rightarrow 1$ | $d \rightarrow 1$ | $d \rightarrow 10$ | $d \rightarrow 10$ | $d \rightarrow 1$ |

For all experiments, we use the same NFM backbone without a single architectural modification and equip it with a task-specific feature-to-feature linear projection. The hyper-parameter settings (empirically chosen) are specialized for each task as shown in **Table 5**.

## D.4 EXPERIMENTAL SETUP: FORECASTING

Long-term times-series forecasting task ($m_f = 1$ and $m_\tau > 1$) is conducted on 7 benchmark datasets (Zhou et al., 2021), including 4 *ETTs* (7 channels), *Weather* (21 channels), *Electricity* (321 channels), and *Traffic* (862 channels) datasets. We use a lookback window of 720 ($N = 720$) for all datasets except for ETTh1 for which $N = 360$ in NFM to make prediction over 4 horizons $\{96, 192, 360, 720\}$. For data setup, see **Appendix D.8**.

We adopt the canonical modelling strategy (Xu et al., 2023; Zeng et al., 2023; Nie et al., 2022) of 1) channel-independence modelling and 2) a simple normalization trick to inputs and the final outputs for dealing with distribution shift caused by non-stationarity in complex time series data. For the latter, we adopt a recent popular choice of it, Reversible Instance Normalization (RevIN) (Kim et al., 2021). Meanwhile, in NFM, we also found that the normalization trick with only mean statistics works far better than with RevIN for some datasets, leading to more stable training and no early saturation to low performance regime. We use only mean statistics as the normalization trick in forecasting on ETTm2 and ETTh2 datasets.

**Forecasting baseline results.** We compare the result of NFM with that of 5 SOTA forecasting models, including FITS (Xu et al., 2023), N-Linear (Zeng et al., 2023), iTransformer (Liu et al., 2023), PatchTST (Nie et al., 2022), TimesNet (Wu et al., 2022). For fair comparison, we collect the best performed results (of single prediction head but not channel-wise prediction heads) and adopted them after conducting a confirmation experiment using their official implementation (FITS[1], PatchTST[2], and N-Linear[3]). We replace them with our result if the difference was over $\pm 5\%$. Note that, as different models will have different regime for optimal lookback windows (although in general, sufficiently longer lookback would yield better results with larger the receptive field), we do not necessarily equalize the length of lookback window, and accordingly we do not compare the performance of the models with respect to different lookback window. For iTransformer[4], we report new forecasting results made on the lookback window of 720 as the originally reported results were made with the lookback window of 96 and the performance of iTransformer is better in this setting. For TimesNet[5], we use the lookback window of 96 (same as the original work) as we observe

---

[1] https://anonymous.4open.science/r/FITS

[2] https://github.com/yuqinie98/PatchTST

[3] https://github.com/cure-lab/LTSF-Linear

[4] https://github.com/thuml/iTransformer

[5] https://github.com/thuml/timesnet

consistently better performance with 96 over 720. Besides, we account only for the main results of the forecasting models made with "a single predictor" instead of channel-wise predictors for a fair comparison and due to its impracticality. For FITS, we use the hyperparameters that the official work used for their final results but **not 10K parameter setup** (cut-off frequency and output frequency of around 100), because with which the performance of FITS is no longer comparable with SOTAs.

**Forecasting baseline results at different sampling rate.** Regarding the experiments of forecasting with different sampling rate in **Section 4.2**, surprisingly, none of the forecasting baseline models can deal with varying-length time series inputs. This is due to their prediction head whose parameters are subject to the initial input length, thus not allowing the estimation directly from downsampled input sequences. While this can be overcome in FITS - by having downsampled factors up to the training cut-off frequency or by posing a lower cut-off frequency during testing time, this task is not directly applicable to PatchTST and N-Linear that working in time domain. A common practice to this is to resample the downsampled input sequences to match the resolution back (i.e., by upsampling). We simply do this by interpolating the downsampled input sequences through zero-padding them in frequency domain (equivalent to applying a sinc kernel in time domain). Note that the resolution of the target prediction remains unchanged (e.g., the case of forecasting on ETTm1 with SR=$1/4$ can be seen as predicting ETTm1 from ETTh1).

### D.5    EXPERIMENTAL SETUP: CLASSIFICATION

For classification, we follow the same experimental setup used in CKconv (Romero et al., 2021) and S4 (Gu et al., 2021). For data setup, see **Appendix D.8**.

**Classification baseline results.** Learning a function-to-function mapping given discrete signals is a desirable property for models to generalize models across different challenging scenarios of time series analysis. Several works have addressed it, modelling continuously evolving dynamics in hidden states (Chen et al., 2018; Rubanova et al., 2019; Morrill et al., 2021; De Brouwer et al., 2019) and a stochastic mapping (Ziegler & Rush, 2019; Deng et al., 2020). On the other hand, this property can also be achieved by simply modelling data directly in Fourier domain with the fact that the DFT provides a function of frequency as samples of a new "functional" representation of the time-domain discrete signals. Intuitively, working explicitly with the frequency samples of discrete signals can be equivalently seen as working with their continuous-time elements in function space. To show this (especially, in the scenario of testing at different sampling rate), we compare the results of NFM with 8 baseline models (7 continuous-time models and 1 Transformer), including ODE-RNN (Rubanova et al., 2019), GRU-$\Delta t$ (Kidger et al., 2020), GRU-ODE (De Brouwer et al., 2019), NCDE (Kidger et al., 2020), NRDE (Morrill et al., 2021)), S4, CKConv, and vanilla Transformer (Vaswani et al., 2017). The results were collected and verified using the implementations in CKconv[6] and S4[7].

### D.6    EXPERIMENTAL SETUP: ANOMALY DETECTION

We employ 4 popular anomaly detection benchmark datasets, including *SMD* (Server Machine Dataset, (Su et al., 2019)), *PSM* (Pooled Server Metrics, (Abdulaal et al., 2021)), *MSL* (Mars Science Laboratory rover, (Su et al., 2019)) *SMAP* (Soil Moisture Active Passive satellite, (Su et al., 2019)), and follow the well-established experimental protocol in (Shen et al., 2020) and the evaluation methodology in (Xu et al., 2021a). We use the same setup used in the work (Xu et al., 2021a) - a window of length 100 for all datasets and baselines, and anomaly ratio (%) of 1.0 (MSL), 1.0 (PSM), 1.0 (SMAP), and 0.5 (SMD). Besides, since we workin in physical space of the time series, we apply channel-independent modelling and input normalization trick just like in forecasting task. For data setup, see **Appendix D.8**.

---

[6]https://github.com/dwromero/ckconv
[7]https://github.com/state-spaces/s4/tree/main

**Anomaly detection baseline results.** We compare the results of NFM with 6 SOTA baselines, including vanilla Transformer (Vaswani et al., 2017), PatchTST (Nie et al., 2022), TimesNet (Wu et al., 2022), ADformer (Xu et al., 2021a), N-linear (Zeng et al., 2023), and FITS (Xu et al., 2023). For the baseline results, we follow the same evaluation methodology used in (Wu et al., 2022; Xu et al., 2021a) and produced the results, using the implementations in TimesNet, ADformer[8], and FITS. Meanwhile, it is important to point out that there were some flaws (in training/validation/training data assignment, computing anomaly threshold, and estimation) in their (all three) official experimental codes that affect the final results. The fixed code samples are available in our repository, and all results in anomaly detection were made again with the fixed codes.

## D.7    OPTIMIZATIONS

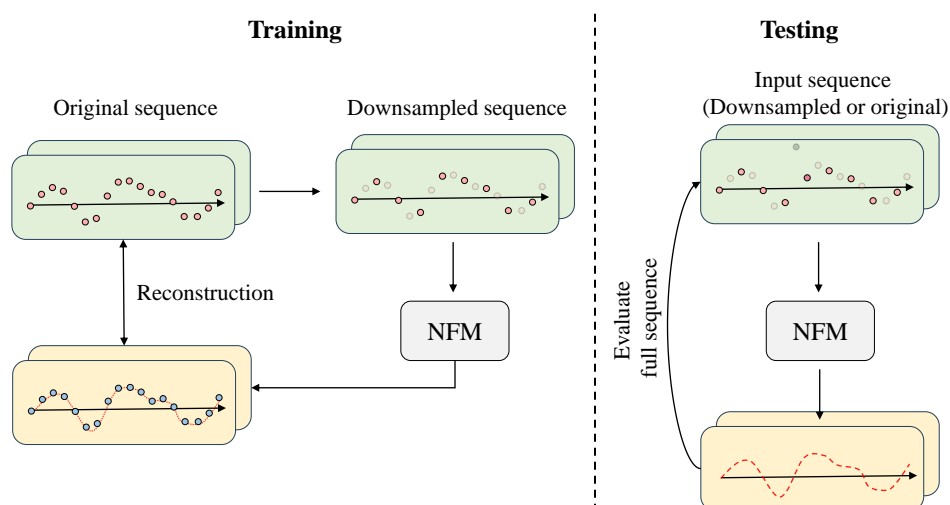

Figure 9: Training and testing framework for anomaly detection task in NFM.

**Forecasting.** Given an input lookback sequence of length $N$, $x[n \in I_N]$ ($f_x = N$ and $T_x = 1$), the aim in forecasting task is to predict a future horizon of a desired length. While most of existing models (Zeng et al., 2023; Liu et al., 2023; Nie et al., 2022; Wu et al., 2022) directly outputs only the target horizon given the input lookback sequence, NFM yields a whole extrapolated sequence (frequency-domain interpolation with $m_t = L/N$) of length $L = N + horizon$, $\hat{y}[n \in I_L]$ ($f_y = f_x$ and $T_y = T_x + \frac{horizon}{f_y}$). During training, we supervise NFM over the whole extrapolated sequence in both time domain and frequency domain against its ground truth $y[n \in I_L] = \{x[N \in I_N], x[N], \ldots, x[L-1]\}$. Note that we observe that NFM in forecasting task performs consistently better when optimized in both time domain and frequency domain. We opt for the standard time-domain loss function $\mathcal{L}_{TD}$, **MSE**, and for the frequency domain loss function $\mathcal{L}_{FD}$ we modify the **focal frequency loss** (Xie et al., 2022; Jiang et al., 2021) used for the recovery of image spectrum and adapt it for time series.

---

[8]https://github.com/thuml/Anomaly-Transformer

$$\mathcal{L}_{Forecasting} = \underbrace{\lambda \frac{1}{L} \sum_{n=0}^{L-1} ||\hat{y}[n] - y[n]||_2}_{\mathcal{L}_{TD}}$$

$$\underbrace{+ (1-\lambda) \frac{1}{K_L} \sum_{k=0}^{K_L-1} ((\hat{Y}_{Real}[k] - Y_{Real}[k])^2 + (\hat{Y}_{Imag}[k] - Y_{Imag}[k])^2)^{1/2}}_{\mathcal{L}_{FD}} \tag{13}$$

where $Y_{Real}$ and $Y_{Imag}$ are the real and imaginary part of the frequency representation of the sequence $y$, and $\lambda$ (we set $\lambda = 0.5$) controls the contribution of the frequency-domain loss. As seen, the distance metric in $\mathcal{L}_{FD}$ considers both amplitude (the contribution of each frequency component to the time-domain signal) and phase (temporal delay introduced by each frequency component) information by operating on both real and imaginary parts of the complex frequency representation. Importantly, one can see that the time-domain objective is local as applied point wise and the frequency-domain objective is global with the fact that each frequency component is a summary about the entirety of the sequence at different periods. Hence, this encourages more faithful construction for the extrapolated sequence as a whole. We argue that this is a reason for NFM working the best when both domains' objective is incorporated.

**Classification.** In classification setup, given pairs of the sequence $x$ of length $N$ and the class label $y \in \{1, \ldots, K\}$, the NFM backbone yields the latent features $z[n \in I_L] = y(n/f_y)$, that resides on the same timespan ($T_y = T_x$) as $x$, with respect to the class information. We train the NFM backbone with a linear classifier (global average pooling + a fully-connected layer) and optimize them using the standard classification loss function, **Cross Entropy**.

**Anomaly detection.** In anomaly detection task, the aim of NFM is to learn the dominant contexts (i.e., normal contexts) of the input sequences in unsupervised manner (i.e., no anomaly label is available). Then, during testing time, the learned sequence-wise normal contexts are used as a standard to establish a decision boundary for anomaly, and the elements of the sequences are evaluated within the corresponding contexts. To achieve this, we frame the objective of NFM as a context learning (see **Figure 8**) and train NFM to learn as faithful contexts as possible. More specifically, given the sequence $x[n \in I_N]$, we downsample (at equidistant sampling rate against the original discretization) it by a downsampling factor $dr$. Denoting the downsampled sequence $x_d[n \in I_{N_d}]$ where $N_d = N/dr$ (i.e., $f_{x_d} = f_x/dr$ and $m_f = dr$), the NFM takes in $x_d$ as input and is trained to reconstruct the full original sequence $\hat{x}[n \in I_N]$ on both **MSE loss** ($\mathcal{L}_{TD}$) and the **focal frequency loss** ($\mathcal{L}_{FD}$) as follows:

$$\mathcal{L}_{AD} = \underbrace{\lambda \frac{1}{N} \sum_{n=0}^{N-1} ||\hat{x}[n] - x[n]||_2}_{\mathcal{L}_{TD}}$$

$$\underbrace{+ (1-\lambda) \frac{1}{K_N} \sum_{k=0}^{K_N-1} ((\hat{X}_{Real}[k] - X_{Real}[k])^2 + (\hat{X}_{Imag}[k] - X_{Imag}[k])^2)^{1/2}}_{\mathcal{L}_{FD}} \tag{14}$$

We set $\lambda = 0.5$, and the $\mathcal{L}_{FD}$ is only applied during training time and not used in any steps of anomaly detection. Besides, it is noteworthy that during the testing time, inputs can be any of original or downsampled version of candidate sequence with resolution-invariance property of NFM, and the evaluation of the sequence points for normality is made on the full length between the original sequence and the restored sequence of the input candidate. Additionally, we note that NFM requires no single architectural modification to adopt the above formulation or changing the above formulation to full reconstruction.

## D.8 SUMMARY OF DATASETS

Table 6: Summary of data settings. SC: SpeechCommand, AD: Anomaly detection, and CLS: Classification.

| Tasks | Dataset | channels | Train / Val / Test | N / L | Num.Class | Domain |
|---|---|---|---|---|---|---|
| Forecasting | ETTm1 | 7 | 60%/20%/20% | (720, 720, 720, 720) / (816, 912, 1056, 1480) | - | Temperature |
| | ETTm2 | 7 | 60%/20%/20% | (720, 720, 720, 720) / (816, 912, 1056, 1480) | - | Temperature |
| | ETTh1 | 7 | 60%/20%/20% | (360, 360, 360, 360) / (452, 552, 696, 1080) | - | Temperature |
| | ETTh2 | 7 | 60%/20%/20% | (720, 720, 720, 720) / (816, 912, 1056, 1480) | - | Temperature |
| | Weather | 21 | 70%/20%/10% | (720, 720, 720, 720) / (816, 912, 1056, 1480) | - | Weather |
| | Electricity | 321 | 70%/20%/10% | (720, 720, 720, 720) / (816, 912, 1056, 1480) | - | Electricity |
| | Traffic | 862 | 70%/20%/10% | (720, 720, 720, 720) / (816, 912, 1056, 1480) | - | Transportation |
| AD | SMD | 38 | 80%/20%/ - | 50/100 | 2 | Server Machine |
| | MSL | 55 | 80%/20%/ - | 50/100 | 2 | Spacecraft |
| | SMAP | 25 | 80%/20%/ - | 50/100 | 2 | Spacecraft |
| | PSM | 25 | 80%/20%/ - | 50/100 | 2 | Server Machine |
| CLS | SC-raw | 1 | 70%/15%/15% | 16000/ - | 10 | Speech |
| | SC-MFCC | 20 | 70%/15%/15% | 161/ - | 10 | Speech |

# E APPENDIX: ADDITIONAL EXPERIMENTS AND ANALYSIS

Here, we provide extra experimental results and analysis omitted in the main work due to the limited work space.

## E.1 FULL FORECASTING RESULTS

**Discussion on the number of parameters.** With the prevalence of chunk-to-chunk prediction in the forecasting community, one practice in the deep forecasting baselines is to **adopt a prediction head that acts on temporal dimension**. Note that the linear models (FITS and N-Linear) naturally fall in this as they by themselves are the prediction head operating on the input time series. Due to this, they scale poorly with the length of horizons as well as the length of lookback window. For example, more than 95% of the learnable weights in PatchTST for $L = 720$ surprisingly belongs to the single "wide" prediction head of 8.3M parameters against 0.4M parameters in its Transformer-based backbone. In contrast, the prediction head used in NFM is feature-to-feature projection and the number of parameters in NFM is *completely independent* from the length of input sequence and prediction horizon. Importantly, we highlight that this aspect completely decouples the contribution of the linear head in modelling sequence and further validates the effectiveness of NFM to modelling temporal dependency unlike the other deep forecasting baselines that are equipped with a **wide** linear predictor.

Table 7: Full long-term forecasting results, where the best results are in **bold** and the second best are underlined. The number of parameters of the baselines are computed based on their original hyper-parameter setting.

| | | NFM (27K) | | FITS (∼0.2M) | | N-Linear (∼0.5M) | | iTransformer (∼5.3M) | | PatchTST (∼8.7M) | | TimesNet (∼0.3B) | |
|---|---|---|---|---|---|---|---|---|---|---|---|---|---|
| | | MSE | MAE | MSE | MAE | MSE | MAE | MSE | MAE | MSE | MAE | MSE | MAE |
| ETTm1 | 96 | **0.286** | **0.338** | 0.309 | 0.352 | 0.306 | 0.348 | 0.319 | 0.367 | 0.293 | 0.346 | 0.338 | 0.375 |
| | 192 | **0.326** | **0.364** | 0.338 | 0.369 | 0.349 | 0.375 | 0.347 | 0.389 | 0.333 | 0.370 | 0.374 | 0.387 |
| | 336 | **0.362** | **0.384** | 0.366 | 0.385 | 0.375 | 0.388 | 0.382 | 0.409 | 0.369 | 0.392 | 0.410 | 0.411 |
| | 720 | **0.406** | 0.414 | 0.415 | **0.412** | 0.433 | 0.422 | 0.437 | 0.440 | 0.416 | 0.420 | 0.478 | 0.450 |
| ETTm2 | 96 | **0.160** | **0.250** | 0.163 | 0.254 | 0.167 | 0.255 | 0.180 | 0.274 | 0.166 | 0.256 | 0.187 | 0.267 |
| | 192 | 0.221 | **0.291** | **0.217** | **0.291** | 0.221 | 0.293 | 0.243 | 0.316 | 0.223 | 0.296 | 0.249 | 0.309 |
| | 336 | 0.271 | **0.326** | **0.268** | **0.326** | 0.274 | 0.327 | 0.299 | 0.352 | 0.274 | 0.329 | 0.321 | 0.351 |
| | 720 | **0.349** | **0.378** | **0.349** | 0.379 | 0.368 | 0.384 | 0.382 | 0.404 | 0.362 | 0.385 | 0.408 | 0.403 |
| ETTh1 | 96 | **0.363** | **0.389** | 0.372 | 0.395 | 0.374 | 0.394 | 0.392 | 0.423 | 0.370 | 0.400 | 0.384 | 0.402 |
| | 192 | **0.404** | **0.413** | **0.404** | **0.413** | 0.408 | 0.415 | 0.428 | 0.447 | 0.413 | 0.429 | 0.436 | 0.429 |
| | 336 | **0.420** | **0.422** | 0.427 | 0.427 | 0.429 | 0.427 | 0.494 | 0.488 | 0.422 | 0.440 | 0.491 | 0.469 |
| | 720 | 0.442 | 0.457 | **0.424** | **0.446** | 0.440 | 0.453 | 0.699 | 0.606 | 0.447 | 0.468 | 0.521 | 0.500 |
| ETTh2 | 96 | 0.281 | 0.341 | **0.271** | **0.337** | 0.277 | 0.338 | 0.304 | 0.364 | 0.274 | **0.337** | 0.340 | 0.374 |
| | 192 | 0.350 | 0.387 | **0.332** | **0.374** | 0.344 | 0.381 | 0.432 | 0.435 | 0.341 | 0.382 | 0.402 | 0.414 |
| | 336 | 0.377 | 0.416 | 0.354 | 0.395 | 0.357 | 0.400 | 0.443 | 0.451 | **0.329** | **0.384** | 0.452 | 0.452 |
| | 720 | 0.414 | 0.455 | **0.378** | 0.423 | 0.394 | 0.436 | 0.441 | 0.469 | 0.379 | **0.422** | 0.462 | 0.468 |
| Weather | 96 | 0.154 | 0.203 | 0.169 | 0.224 | 0.182 | 0.232 | 0.173 | 0.227 | **0.149** | **0.198** | 0.172 | 0.220 |
| | 192 | 0.198 | 0.246 | 0.213 | 0.261 | 0.225 | 0.269 | 0.219 | 0.262 | **0.194** | **0.241** | 0.219 | 0.261 |
| | 336 | **0.245** | **0.281** | 0.259 | 0.296 | 0.271 | 0.301 | 0.283 | 0.310 | **0.245** | 0.282 | 0.280 | 0.306 |
| | 720 | **0.312** | **0.331** | 0.321 | 0.340 | 0.338 | 0.348 | 0.344 | 0.355 | 0.314 | 0.334 | 0.365 | 0.359 |
| Electricity | 96 | 0.131 | **0.222** | 0.135 | 0.231 | 0.141 | 0.237 | 0.132 | 0.227 | **0.129** | **0.222** | 0.168 | 0.272 |
| | 192 | **0.147** | **0.240** | 0.149 | 0.244 | 0.154 | 0.248 | 0.155 | 0.252 | **0.147** | **0.240** | 0.184 | 0.289 |
| | 336 | **0.163** | **0.256** | 0.165 | 0.261 | 0.171 | 0.265 | 0.170 | 0.267 | **0.163** | 0.259 | 0.198 | 0.300 |
| | 720 | **0.194** | **0.284** | 0.203 | 0.293 | 0.210 | 0.297 | 0.195 | 0.289 | 0.197 | 0.290 | 0.220 | 0.320 |
| Traffic | 96 | 0.367 | **0.249** | 0.386 | 0.269 | 0.410 | 0.279 | **0.344** | 0.254 | 0.360 | **0.249** | 0.593 | 0.321 |
| | 192 | 0.377 | **0.252** | 0.398 | 0.274 | 0.423 | 0.284 | **0.366** | 0.265 | 0.379 | 0.256 | 0.617 | 0.336 |
| | 336 | 0.392 | **0.259** | 0.410 | 0.278 | 0.435 | 0.290 | **0.381** | 0.273 | 0.392 | 0.264 | 0.629 | 0.336 |
| | 720 | 0.427 | **0.279** | 0.448 | 0.296 | 0.464 | 0.307 | **0.413** | **0.287** | 0.432 | 0.286 | 0.640 | 0.350 |

## E.2 FULL ANOMALY RESULTS

Table 8: Time series anomaly detection results on 4 datasets. The higher the three metrics, including the precision (P), recall (R), and F1-score (F1) in percentage, are, better the performance.

| Model (params) | SMD | | | MSL | | | SMAP | | | PSM | | |
|---|---|---|---|---|---|---|---|---|---|---|---|---|
| | P | R | F1 | P | R | F1 | P | R | F1 | P | R | F1 |
| Transformer (0.2M) | 68.40 | 85.37 | 75.95 | 88.11 | 76.55 | 81.93 | 89.37 | 57.12 | 69.70 | 99.96 | 79.80 | 88.75 |
| PatchTST (0.2M) | 80.33 | 83.96 | 82.11 | 84.33 | 77.01 | 80.51 | **92.22** | 55.27 | 69.11 | 98.78 | 93.38 | 96.00 |
| TimesNet (∼28M) | 82.67 | 83.88 | 83.27 | 87.45 | 76.65 | 81.70 | 89.07 | **62.17** | **73.23** | 98.42 | 96.20 | 97.30 |
| ADformer*(4.8M) | 68.79 | **85.86** | 76.38 | 88.68 | 75.87 | 81.78 | 91.85 | 58.11 | 71.18 | **99.98** | 71.15 | 83.14 |
| N-Linear (10K) | 78.94 | 84.89 | 81.81 | 86.55 | 76.43 | 81.18 | 89.85 | 54.05 | 67.50 | 98.47 | 93.22 | 95.77 |
| FITS (1.3K) | 79.90 | 83.52 | 81.67 | 86.85 | 75.49 | 80.77 | 88.47 | 50.22 | 64.07 | 98.74 | 94.55 | 96.60 |
| **NFM** (6.6K) | **86.82** | 81.96 | **84.32** | **88.72** | **77.11** | **82.46** | 90.12 | 58.87 | 70.88 | 98.92 | **96.91** | **97.51** |

* The joint criterion in ADformer is replaced with the simple reconstruction error to compute anomaly score for fair comparison.

## E.3 FULL RESULTS OF FORECASTING AT DIFFERENT INPUT RESOLUTION

In practice, it is not rare to encounter a scenario where a system undergoes or necessitates a change in sampling rate for signals being monitored by a model. Such change does not affect the underlying temporal dynamic of the signals (i.e., the same solution) but brings in a positional alternation in the sequences of our observations, greatly affecting the performance of the model. To this end, we conduct forecasting on the input time series sampled at "unseen" discretization rate (this experiment is made for the first time in our work). Overall, the full results in **Table 9** demonstrates the resolution-invariance property of NFM that can be highly valuable in practical applications.

Table 9: Full forecasting results (MSE) and performance drops (%) at different testing-time sampling rate, where SR$= f_x^{test}/f_x^{train}$. The best performance is in **blue** and the least performance drop in **red**.

| Dataset | Horizon | NFM | | FITS | | N-Linear | | PatchTST | |
|---|---|---|---|---|---|---|---|---|---|
| | | 1/4 | 1/6 | 1/4 | 1/6 | 1/4 | 1/6 | 1/4 | 1/6 |
| ETTm1 | 96 | **0.299** (**4.5** ↓) | **0.319** (**11.5** ↓) | 0.348 (12.6 ↓) | 0.368 (19.1 ↓) | 0.436 (42.5 ↓) | 0.482 (57.5 ↓) | 0.394 (34.5 ↓) | 0.434 (48.5 ↓) |
| | 192 | **0.339** (**4.0** ↓) | **0.349** (**7.1** ↓) | 0.363 (7.4 ↓) | 0.377 (11.5 ↓) | 0.444 (27.2 ↓) | 0.481 (37.8 ↓) | 0.386 (15.9 ↓) | 0.412 (23.7 ↓) |
| | 336 | **0.363** (**0.3** ↓) | **0.369** (**2.1** ↓) | 0.384 (4.9 ↓) | 0.392 (7.1 ↓) | 0.457 (21.9 ↓) | 0.497 (32.5 ↓) | 0.394 (6.8 ↓) | 0.412 (11.7 ↓) |
| | 720 | **0.407** (**0.2** ↓) | **0.414** (**2.0** ↓) | 0.426 (2.9 ↓) | 0.437 (5.3 ↓) | 0.469 (8.3 ↓) | 0.471 (8.8 ↓) | 0.433 (4.1 ↓) | 0.443 (6.5 ↓) |
| ETTm2 | 96 | **0.179** (**11.9** ↓) | **0.189** (**18.2** ↓) | 0.198 (21.5 ↓) | 0.216 (31.7 ↓) | 0.251 (50.3 ↓) | 0.281 (68.3 ↓) | 0.232 (39.8 ↓) | 0.265 (59.6 ↓) |
| | 192 | **0.230** (**4.1** ↓) | **0.239** (**8.1** ↓) | 0.240 (10.6 ↓) | 0.258 (16.2 ↓) | 0.267 (20.8 ↓) | 0.284 (28.5 ↓) | 0.264 (18.4 ↓) | 0.287 (28.7 ↓) |
| | 336 | **0.281** (**3.7** ↓) | **0.287** (**6.0** ↓) | 0.288 (7.5 ↓) | 0.300 (11.9 ↓) | 0.307 (12.0 ↓) | 0.331 (20.8 ↓) | 0.297 (8.4 ↓) | 0.311 (13.5 ↓) |
| | 720 | **0.356** (**2.0** ↓) | **0.358** (**2.6** ↓) | **0.356** (**2.0** ↓) | 0.364 (4.3 ↓) | 0.409 (11.1 ↓) | 0.422 (14.7 ↓) | 0.382 (5.5 ↓) | 0.396 (9.4 ↓) |
| Weather | 96 | **0.164** (**6.5** ↓) | **0.173** (**12.3** ↓) | 0.182 (7.7 ↓) | 0.195 (14.8 ↓) | 0.268 (47.3 ↓) | 0.280 (53.8 ↓) | 0.212 (42.3 ↓) | 0.236 (58.4 ↓) |
| | 192 | **0.209** (5.6 ↓) | **0.219** (10.6 ↓) | 0.222 (**4.2** ↓) | 0.233 (**9.4** ↓) | 0.271 (20.4 ↓) | 0.288 (28.0 ↓) | 0.261 (34.5 ↓) | 0.282 (45.4 ↓) |
| | 336 | **0.253** (3.3 ↓) | **0.258** (5.3 ↓) | 0.265 (**2.3** ↓) | 0.272 (**5.0** ↓) | 0.296 (9.2 ↓) | 0.308 (13.7 ↓) | 0.278 (13.5 ↓) | 0.292 (19.2 ↓) |
| | 720 | **0.315** (**1.0** ↓) | **0.318** (**1.9** ↓) | 0.325 (1.2 ↓) | 0.329 (2.5 ↓) | 0.351 (3.8 ↓) | 0.364 (7.7 ↓) | 0.329 (4.8 ↓) | 0.335 (6.7 ↓) |

**Analysis.** Especially, we observe that the models operating fully on frequency domain (NFM and FITS) are much more robust to the change in sampling rate as learning a function-to-function mapping, than those operating on time domain (PatchTST and N-Linear). Interestingly, the performance degradation with unseen sampling rate tends to be more significant in forecasting over short horizons (with the same lookback length) and relatively minor over long horizons in all models. This tendency could be a strong indication that the forecasting models, including NFM, become more reliant on the global context (or similarly, low frequency regime) of the lookback window and less focusing on the local details (or similarly, high frequency regime) in the window as the predicting horizon gets longer. In this sense, the results imply that NFM is not the one that leverages local features in optimal way but is less prone to the local variations than the others. In the future, integrating a mechanism that encourages learning more of local features (high-frequency information) into models could potentially improve the forecasting ability in the long horizon cases as well as the short horizon cases.

### E.4 FULL TABULAR RESULTS ON COMPARISON OF NFM WITH DIFFERENT ABLATION CASES

Table 10: Tabular results of the ablation study on ETTm1. We use the same set up used in **Table 5** for all cases and the number of heads = 2 for AFNO and AFF.

| Horizon | Metric | INFF | | | LFT | | | |
|---|---|---|---|---|---|---|---|---|
| | | × | Naive | **LFT** | FNO | AFNO | GFN | AFF |
| 96 | MSE | 0.296 | 0.292 | 0.286 | 0.289 | 0.291 | 0.303 | 0.292 |
| | MAE | 0.348 | 0.343 | 0.338 | 0.347 | 0.347 | 0.349 | 0.346 |
| | FLOP (G) | 0.076 | 0.076 | 0.082 | 0.069 | 0.090 | 0.066 | 0.090 |
| | PMU (GB) | 0.029 | 0.030 | 0.030 | 0.030 | 0.032 | 0.025 | 0.033 |
| | Params (M) | 0.021 | 0.036 | 0.027 | 0.435 | 0.020 | 0.029 | 0.020 |
| 192 | MSE | 0.335 | 0.330 | 0.326 | 0.330 | 0.338 | 0.338 | 0.340 |
| | MAE | 0.370 | 0.364 | 0.364 | 0.367 | 0.374 | 0.372 | 0.372 |
| | FLOP (G) | 0.085 | 0.085 | 0.089 | 0.080 | 0.099 | 0.072 | 0.099 |
| | PMU (GB) | 0.031 | 0.033 | 0.033 | 0.033 | 0.035 | 0.027 | 0.036 |
| | Params (M) | 0.021 | 0.039 | 0.027 | 0.484 | 0.020 | 0.030 | 0.020 |
| 336 | MSE | 0.372 | 0.366 | 0.362 | 0.365 | 0.371 | 0.383 | 0.369 |
| | MAE | 0.396 | 0.387 | 0.384 | 0.390 | 0.391 | 0.394 | 0.393 |
| | FLOP (G) | 0.096 | 0.096 | 0.101 | 0.089 | 0.112 | 0.081 | 0.113 |
| | PMU (GB) | 0.036 | 0.036 | 0.037 | 0.038 | 0.039 | 0.030 | 0.040 |
| | Params (M) | 0.021 | 0.039 | 0.027 | 0.557 | 0.020 | 0.033 | 0.020 |
| 720 | MSE | 0.421 | 0.421 | 0.406 | 0.431 | 0.413 | 0.427 | 0.407 |
| | MAE | 0.422 | 0.419 | 0.414 | 0.424 | 0.414 | 0.420 | 0.421 |
| | FLOP (G) | 0.119 | 0.119 | 0.128 | 0.115 | 0.146 | 0.105 | 0.147 |
| | PMU (GB) | 0.047 | 0.048 | 0.048 | 0.051 | 0.038 | 0.051 | 0.053 |
| | Params (M) | 0.021 | 0.043 | 0.027 | 0.557 | 0.020 | 0.039 | 0.020 |

Table 11: Tabular results of the ablation study on SpeechCommand. We use the same set up used in **Table 5** for all cases and the number of heads = 2 for AFNO and AFF.

| SR | Metric | INFF | | | LFT | | | |
|---|---|---|---|---|---|---|---|---|
| | | × | Naive | **LFT** | FNO | AFNO | GFN | AFF |
| 1.0 | ACC (%) | 79.4 | 82.6 | 90.9 | 84.8 | 84.7 | 86.2 | 88.4 |
| 0.5 | ACC (%) | 74.1 | 78.6 | 90.4 | 75.0 | 82.4 | 78.0 | 85.8 |
| | FLOP (G) | 0.480 | 0.480 | 0.487 | 0.383 | 0.478 | 0.330 | 0.478 |
| | PMU (GB) | 0.183 | 0.183 | 0.188 | 0.395 | 0.175 | 0.135 | 0.178 |
| | Params (M) | 0.031 | 0.286 | 0.035 | 16.4 | 0.031 | 0.533 | 0.031 |

### E.5 STATISTICAL SIGNIFICANCE ON THE MAIN RESULTS WITH DIFFERENT RANDOM SEEDS

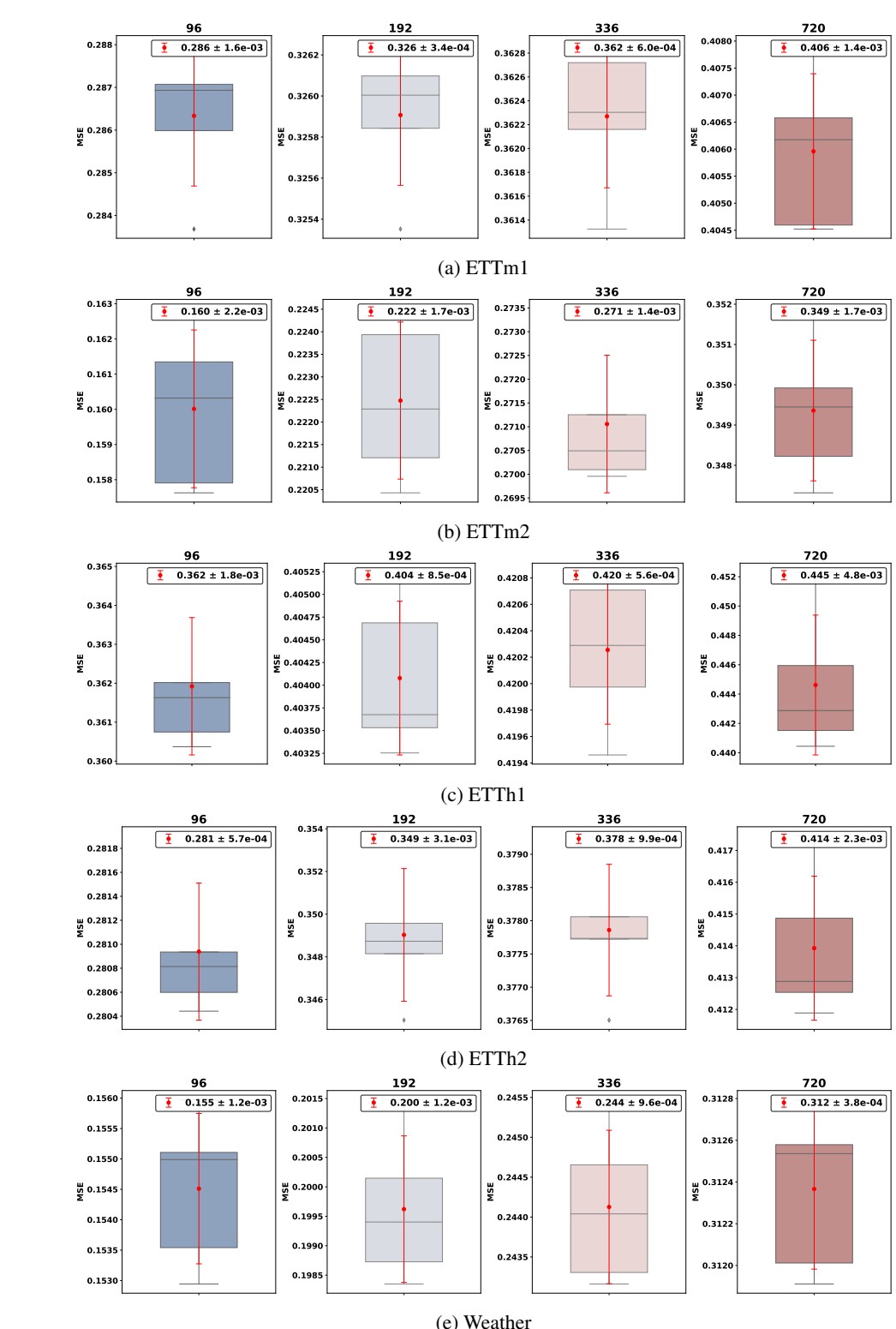

(a) ETTm1

(b) ETTm2

(c) ETTh1

(d) ETTh2

(e) Weather

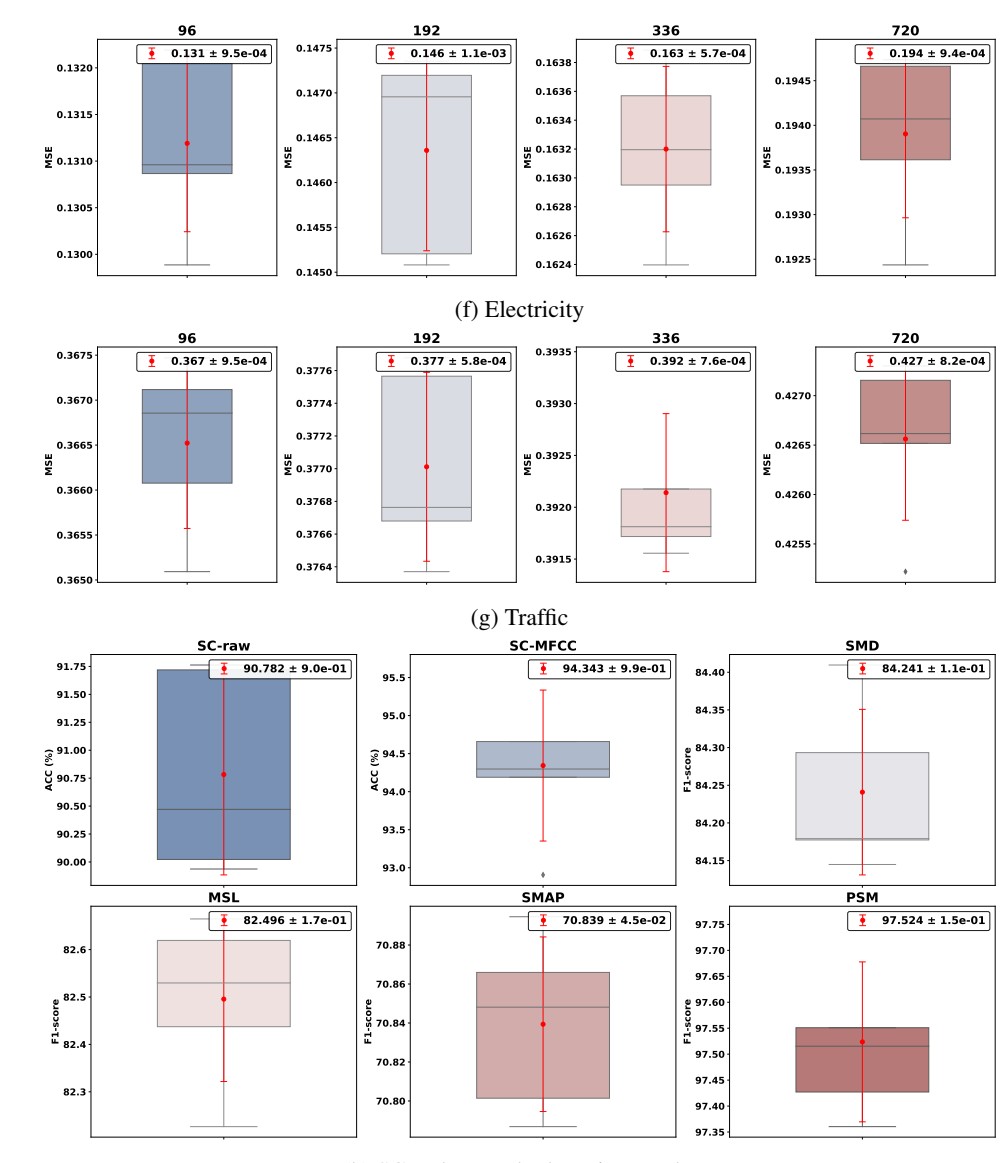

(f) Electricity

(g) Traffic

(h) SC and anomaly detection results

Figure 10: Statistical significance on the main experimental results computed by repeating the main experiments with different random seeds. The number in the title of forecasting results (a) ∼ (g) indicates prediction horizon, and the legend in each box plot $mean \pm std$.

