# OpenReview forum: "Neural Fourier Modelling: A Highly Compact Approach to Time-Series Analysis"
_ICLR.cc/2025/Conference — ICLR 2025 Conference Desk Rejected Submission_

### Official Review · Reviewer_xBs2 · 2024-10-20

**Soundness:** 4
**Presentation:** 4
**Contribution:** 3
**Rating:** 8
**Confidence:** 4

**Summary:**

This paper introduces a neural Fourier modelling (NFM). This approach can model finite-length time series as functions in the Fourier domain and also has the capacity for data manipulation within the Fourier domain. Learnable Frequency Tokens (LFT) and Implicit Neural Fourier Filters (INFF) are two learning modules suggested by the authors to learn NFM. The introduction gives a good motivation for the problem, the literature is vast, and the methods are explained well. The paper shows the efficacy of the proposed approach on different time series tasks (forecasting, anomaly detection, and classification) and compared with other methods.

**Strengths:**

Experiments are vast and thorough.
The paper is written well and the figure provides a clear idea of the approach.

**Weaknesses:**

Since the paper addresses multiple tasks for time series, exhaustive experimentation, and comparisons are a bit lacking.

**Questions:**

It would be interesting to compare performance with FNO (Fourier neural operator). They have been successful on different time series tasks, especially forecasting. A discussion between neural operator and NFM would enhance the paper.
For classification, some non-deep learning SOTA would be great like Minirocket and HIVE. These methods have been very successful for time series classification. Also, adding the benchmarks for classification would enhance the paper even more. Minirocket and other methods and benchmark data information can be found here: https://arxiv.org/pdf/2012.08791

---

### Official Review · Reviewer_VnjZ · 2024-10-28

**Soundness:** 3
**Presentation:** 3
**Contribution:** 3
**Rating:** 6
**Confidence:** 4

**Summary:**

This paper proposes Neural Fourier Machine (NFM), which leverages the FITS features to process multivariate time-series and perform multiple time-series analysis tasks. NFM is a modularized model with two essential components: (1) Learnable Frequency Tokens (LFT) which learns the coefficients of Fourier interpolation/extrapolation, improving the flexibility of Fourier space manipulations, and (2) Implicit Neural Fourier Filter (INFF) which offers more expressive modeling of Fourier features. Experiment results show that NFM achieves comparable performance, and sometimes outperform, to the SOTA baselines in time-series forecasting, anomaly detection, and classification. Additionally, NFM is much smaller in size compared to SOTA deep learning models. Compared to FITS, the scale of NFM is mostly invariant to the dimension of input time-series. Ablation study also demonstrates the effectiveness of proposed components, as well as the scaling of NFM.

**Strengths:**

1. This paper is well-written and easy to follow.
2. The proposed method is a reasonable extension of the FITS representations, leveraging FITS for multivariate time-series and more tasks.
3. Experimental results are comprehensive and demonstrate the strong performance of NFM on various tasks.
4. The ablation study is comprehensive, making it straightforward to assess the effectiveness of the proposed components.

**Weaknesses:**

> There is no major weakness from my perspective, but one concern regarding the choice of classification benchmark:

The authors use the SpeechCommands dataset as the classification benchmark. Although it is a good dataset and a reasonable choice for NFM, audio data is only one of many forms of time series data. Performance on common classification benchmarks, including the UCR and UEA datasets, could provide a more comprehensive assessment of the proposed method, since they contain time series data collected from various sources. If possible, please consider including the performance on the UCR and UEA datasets, or a subset of them.

> Several minor things that may need additional clarification:

1. Figure 2 is somewhat difficult to understand. This figure visualizes the components in the Fourier series in the time domain, where the overlapping series make it difficult to read. The Fourier representations elsewhere in the paper are shown as magnitudes in the frequency domain. Therefore, the "special" visualization in this figure seems unnecessary and could be improved.
2. Abbreviation of inverse DFT: It is defined as IDFT on line 176 and used in the same form in the text, but written as iDFT in Figure 3.
3. The number of parameters in the experiment results can be misleading. For example, in Table 1, NFM has $27K$ parameters and FITS has approximately $0.2M$ parameters. However, in Figure 7, FITS actually has fewer parameters than NFM in many cases. This is because NFM first projects a c-channel time series into d dimensions, which makes it channel-invariant. Therefore, in the tables, the number of parameters for the baseline methods should be a range, such as $20K \sim 0.2M$, instead of taking the maximum of approximately $0.2M$. And there should be an additional note to discuss the source of these numbers and clarify that they could vary across datasets.
4. The scaling case presented in Figure 6(a) may not be the most representative one, since the ETTm1 dataset only has 7 channels, and all the cases have $d > c$. It would be more interesting to present the Traffic dataset, which has over 800 channels.

**Questions:**

1. Why are the common classification benchmark datasets (UCR and UEA datasets) not considered in this paper? If there is a specific reason that makes them inapplicable, could you please explain?
2. As a more general and expressive form of FITS, based on Table 1, NFM outperforms FITS on most of the datasets except ETTh2. Could you provide some insights and explanations on this? Is there anything special about this dataset?
3. Between the modules of NFM, the variables are projected between time and frequency domains with DFT and iDFT. What if everything is kept in the frequency domain, i.e., removing the iDFT? It seems the only operation in the time domain is the channel mixing. What if the channel mixing is also computed with Fourier features?
4. In Table 4, the SR is always smaller than 1, where the sampling rate in the training set ($f_x^{train}$) is always higher than the test set, i.e., the training data contains more details. Would $\text{SR} > 1$ also be a valid setup? How would the NFM perform in such a case?

---

### Official Review · Reviewer_FB7k · 2024-10-29

**Soundness:** 3
**Presentation:** 3
**Contribution:** 3
**Rating:** 6
**Confidence:** 3

**Summary:**

The paper proposed a time series analysis method that leverages the Fourier transform's properties for data manipulation, incorporating frequency extrapolation and interpolation as core learning mechanisms.

**Strengths:**

1. The paper is well-origanized and easy to follow.

2. The paper innovatively shifts the focus of time-series analysis from the time domain to the Fourier domain, providing a new aspect for time series analysis.

3. The authors proposed a method Neural Fourier Modelling (NFM) that effectively utilizes the Fourier transform's properties for data manipulation and the proposed method is able to handle diverse time-series tasks.

**Weaknesses:**

1. The Fourier transform and its inverse have been utilized in the time series forecasting domain, and the proposed Learnable Frequency Tokens (LFT) appear similar to prior works, such as FEDformer [1]. The authors should discuss the differences and strengths of the proposed LFT in comparison to these existing methods.

2. The proposed Implicit Neural Fourier Filters (INFF) are designed to achieve an expressive continuous global convolution for learning interpolation and extrapolation in the Fourier domain. Would it not be beneficial to consider using Frequency Channel Attention [2] for this purpose?

3. Several typos are present in the manuscript and should be corrected to enhance the overall clarity.

[1] FEDformer: Frequency Enhanced Decomposed Transformer for Long-term Series Forecasting
[2] FcaNet: Frequency Channel Attention Networks

**Questions:**

1. How to choose K_N and K_L, are they fixed for all time series?

2. Does this method able to deal with both multi-variate time series and single-variate time series?

---

### Official Review · Reviewer_Yio4 · 2024-11-01

**Soundness:** 3
**Presentation:** 2
**Contribution:** 3
**Rating:** 5
**Confidence:** 4

**Summary:**

This paper presents a novel approach to modeling time series in the Fourier domain using an encoder-decoder architecture. The authors introduce two key components: Learnable Frequency Tokens (LFT) and Implicit Neural Fourier Filters (INFF). The proposed architecture is evaluated across several tasks, including forecasting, classification, and anomaly detection. Additionally, the authors demonstrate the model's effectiveness across different discretization rates, highlighting its versatility.

**Strengths:**

- S1: Exploring the use of neural networks in the frequency domain is interesting and brings several advantages, as highlighted by the authors: neural networks with fewer weights, the possibility of modelling the same time series for different sampling frequencies.
- S2: The experimental results for the three tasks considered are good and appear to improve on the state-of-the-art performance with an architecture that has fewer parameters than baselines.
- S3: The appendices are well documented and provide a much more in-depth understanding of the architecture and processing of supervised tasks.
- S4:  I appreciate the limitation section where authors acknowledge that the current implementation of the model not suitable handling irregular time series

**Weaknesses:**

- W1: The paper, excluding the appendices, is difficult to follow. The contributions and positioning of the work are unclear. Additionally, the architecture is not well-explained, and the model's description could benefit from being reorganized. Crucial parts of the architecture are relegated to the appendices, which hinders understanding.

- W2: The model's architecture heavily relies on Implicit Neural Representations (INRs), particularly in:
    - The input projection block (I don't understand why you apply SIREN to $x$ input in appendix D.2., could you explain?)
    - The LFT embedding
    - The INFF block

INR for time series is an active area of research, with applications in generation [1], forecasting [2], forecasting/imputation [3]. These papers also address the sampling problem in time series and emphasize the advantages of using time-index (frequency-domain) models. It is surprising that the paper does not discuss these related works at all.

- W3: While handling three tasks (forecasting, classification, and anomaly detection) might seem like a strength, it makes the paper feel unfocused. For instance, using a single speech classification dataset (one that favors frequency-domain processing) while comparing against baselines that are not state-of-the-art in classification, undermines the claims of achieving state-of-the-art performance in classification.

- W4: Other limitations of the architecture are not sufficiently addressed. For example, the inability to handle new samples (new channels) during inference or the fact that the architecture in its current form cannot accept co-variates are important drawbacks that should be discussed.


[1] iHyperTime: Interpretable Time Series Generation with Implicit Neural Representations, TMLR 2024

[2] Learning deep time-index models for time series forecasting, ICML 2023

[3] Time Series Continuous Modeling for Imputation and Forecasting with Implicit Neural Representations, TMLR 2024

**Questions:**

Please see weaknesses. I believe that the suggested improvements could significantly enhance the quality of the paper.

---

### Official Review · Reviewer_8cAE · 2024-11-04

**Soundness:** 3
**Presentation:** 4
**Contribution:** 3
**Rating:** 6
**Confidence:** 4

**Summary:**

- The work concerns time-series analysis, such as forecasting, classification, and anomaly detection.
- The proposed method neatly decouples data size and representation size, effectively making the model resolution-invariant.
- The method is motivated and described theoretically and evaluated empirically.

I did not review the appendix in depth.

**Strengths:**

- The work is of high quality and very well-written.
- I am not aware of these ideas being proposed before. The work is transparent on how it differs from FITS.
- Time series forecasting is relevant in many domains and is far from being solved. The proposed method investigates low-parameter and resolution-invariant models, both being exciting research directions.
- The experiments go beyond mere performance metrics.

**Weaknesses:**

1. The related work does not discuss compact models, a central claimed benefit of the proposed model. Similarly, other resolution-invariant approaches should be provided or stated they do not exist.
2. The choice of classification datasets is very limited (i.e., to a single one).
3. The MLP Channel Mixer, MLP Mixer, and Predictor components (e.g., see Fig. 3) are not discussed sufficiently. While they are not the key component being newly proposed, they appear to be highly relevant. For instance, interleaving time (MLP Channel Mixer) and frequency domain (INFF) operations in the Mixer Block might warrant further discussion. For instance, why was the specific order of operations chosen? The ablation study requires expansion to isolate the contributions of these components versus the newly proposed blocks. This would give readers a clearer understanding of where the performance gains are coming from.

#### Minor Comments
- It would be appropriate to cite the MLP-Mixer and/or TimeMixer works since the proposed method heavily builds on them.
- A possible addition to related work: The famous N-BEATS (Oreshkin et al. 2020, Sec. 3.3) was also presented with learning coefficients for a Fourier basis.
- Language: Missing "and" in l. 047. "a" -> "the" in l. 152. Full stop in l. 166. L. 169 "switching" -> "switch". Fig. 3 "Mixer blcok". Missing closing parenthesis in l. 314. ...
- It might be unintentional that the venues in the references are underlined.
- Side note: Since the (I)DFT is just a matrix multiplication, fusing operations in the LFT/INFF blocks might be possible for faster computations.
- Fig. 6: The colors of parameter counts of the d=8 to d=36 cases differ in subfigure (a) vs. (b).
- Unfortunately, some sections of the extensive (!) appendix are not consistently referenced by the main paper and might go unnoticed.
#### References
- Oreshkin, Boris N., Dmitri Carpov, Nicolas Chapados, and Yoshua Bengio. “N-BEATS: Neural Basis Expansion Analysis for Interpretable Time Series Forecasting.” In International Conference on Learning Representations, 2020.
- Wang, Shiyu, Haixu Wu, Xiaoming Shi, Tengge Hu, Huakun Luo, Lintao Ma, James Y. Zhang, and Jun Zhou. “TimeMixer: Decomposable Multiscale Mixing for Time Series Forecasting.” In The Twelfth International Conference on Learning Representations, 2024.

**Questions:**

Note: The most important questions are listed first.

1. How were the baseline models in the long-term forecasting benchmark selected? In particular, why are better models such as TimeMixer (Wang et al. 2024) not shown? Given the method's strengths, such as excellent parameter parsimony, it would be acceptable not to be best-in-class everywhere.
2. See Weakness #3 above.
3. Why is there no residual around the MLP Channel Mixer and MLP Mixer blocks (Fig. 3), but instead around the stack of all mixer blocks? Is it unnecessary?
4. Appendix D1, l. 935: What is $r$?
5. What is meant by "can be learned without a-priori" (l. 262)?
6. Can the LFT be replaced by a single learned vector $V[k]$ when only a single data sampling rate is observed?
7. How is the complex-valued ReLU in INFF (l. 317) defined? $\mathbb C$ is not ordered, so how does one define a maximum operation?

---

### Note · Program_Chairs · 2024-11-12
**Submission Desk Rejected by Program Chairs**

Violating anonymity policy. The linked github repo has an explicit mentioning of the author identity.